# CAN BERT REFRAIN FROM FORGETTING ON SEQUENTIAL TASKS? A PROBING STUDY

**Mingxu Tao[1,2], Yansong Feng[1,3], Dongyan Zhao[1,2]**
[1]Wangxuan Institute of Computer Technology, Peking University, China
[2]Center for Data Science, Peking University, China
[3]The MOE Key Laboratory of Computational Linguistics, Peking University, China
{thomastao, fengyansong, zhaody}@pku.edu.cn

## ABSTRACT

Large pre-trained language models help to achieve state of the art on a variety of natural language processing (NLP) tasks, nevertheless, they still suffer from forgetting when incrementally learning a sequence of tasks. To alleviate this problem, recent works enhance existing models by sparse experience replay and local adaption, which yield satisfactory performance. However, in this paper we find that pre-trained language models like BERT have a potential ability to learn sequentially, even without any sparse memory replay. To verify the ability of BERT to maintain old knowledge, we adopt and re-finetune single-layer probe networks with the parameters of BERT fixed. We investigate the models on two types of NLP tasks, text classification and extractive question answering. Our experiments reveal that BERT can actually generate high quality representations for previously learned tasks in a long term, under extremely sparse replay or even no replay. We further introduce a series of novel methods to interpret the mechanism of forgetting and how memory rehearsal plays a significant role in task incremental learning, which bridges the gap between our new discovery and previous studies about catastrophic forgetting[1].

## 1 INTRODUCTION

Continual Learning aims to obtain knowledge from a stream of data across time (Ring, 1994; Thrun, 1998; Chen & Liu, 2018). As a booming area in continual learning, task-incremental learning requires a model to learn a sequence of tasks, without forgetting previously learned knowledge. It is a practical scene to train models on a stream of tasks sequentially, avoiding to re-train on all existing data exhaustively once a new task arrives. In natural language processing, although many large-scale pre-trained language models (PLMs) have ceaselessly achieved on new records on various benchmarks, they cannot be directly deployed in a task-incremental setting. These models tend to perform poorly on previously seen tasks when learning new ones. For instance, a BERT$_{\text{BASE}}$ model trained sequentially on text classification tasks may not be able to make any correct predictions for the first task after learning new ones, with almost-zero accuracy scores (d'Autume et al., 2019). This phenomenon is known as *catastrophic forgetting* (McCloskey & Cohen, 1989; French, 1999; Rosenstein et al., 2005). Many existing works design novel architectures or components to alleviate the forgetting when learning incrementally (Kirkpatrick et al., 2017; Zenke et al., 2017; Rebuffi et al., 2017; Mallya & Lazebnik, 2018; d'Autume et al., 2019; Pfeiffer et al., 2020; Sun et al., 2020; Geng et al., 2021; Jin et al., 2022; Qin et al., 2022). Among them, d'Autume et al. (2019) find that an NLP model augmented by sparse memory replay can refrain from forgetting to a great extent. Their method randomly samples 100 instances from old tasks for replay, after learning every 10,000 unseen instances. Considering that their method can regain the ability to process previous tasks via merely 100 instances in 4 steps[2], a question comes to our mind: *Whether pre-trained language models like BERT really suffer from forgetting when learning a sequence of tasks?*

---

[1]Code will be released at https://github.com/kobayashikanna01/plms_are_lifelong_learners

[2]With a training batch of size 32, sampling 100 instances means it takes only 4 steps to recover the forgotten knowledge.

Mehta et al. (2021) reveal that, under proper hyper-parameters, models with pre-training can suffer less catastrophic forgetting than models without pre-training. However, in this work, we specifically focus on the frequency of memory replay. We wonder whether the BERT encoder can still maintain knowledge learned from previous tasks as it performs in d'Autume et al. (2019), with an extremely sparse replay frequency or even without replay.

Probing study has become a popular tool to investigate model interpretability (Tenney et al., 2019; Jawahar et al., 2019). For instance, Wu et al. (2022) probe the continual learning ability of a model by comparing the performance of different PLMs trained with different continual learning strategies. In this paper, our main concern is to examine whether PLMs have an intrinsic ability to maintain previously learned knowledge in a long term. We track the encoding ability of BERT for specific tasks in a BERT *before*, *during*, and *after* it learns the corresponding tasks. Comparing the probing results of models trained under different replay frequencies and trained without memory replay, we find that BERT itself can refrain from forgetting when learning a sequence of tasks. This is somewhat contrary to existing studies about catastrophic forgetting, which further motivates us to investigate how the representations of examples from different tasks are organized in the parameter space. Inspired by prior works (Gao et al., 2019; Wang et al., 2020a), we define the representation sub-space of a class as a convex cone, and provide an algorithm to acquire the narrowest solution. With this toolkit in hand, we find that: after learning several tasks without memory replay, the representation sub-spaces of classes from different tasks will overlap with each other. However, the sub-spaces of classes from the same task keep never-overlapping all along. The former explains the catastrophic forgetting in task-incremental learning from a novel viewpoint of representations, while the the latter explains why BERT has a potential to encode prior tasks even without replay.

Our main contributions in this work are:
(1) we conduct a thorough study to quantitatively characterize how the representation ability of a PLM like BERT change when it continuously learns a sequence of tasks. We are the first to track the encoding ability of previously learned tasks in BERT when learning new tasks continuously.
(2) Our findings reveal that BERT can actually maintain its encoding ability for already learned tasks, and has a strong potential to produce high-quality representations for previous tasks in a long term, under an extremely sparse replay or even without memory replay, which is contrary to previous studies.
(3) We further investigate the topological structure of the learned representation sub-space within a task and among different tasks, and find that the forgetting phenomenon can be interpreted into two aspects, the *intra-task forgetting* and *inter-task forgetting* (Section 4), enabling us to explain the contrary between our findings and previous studies.

## 2 BACKGROUND

Following prior work (Biesialska et al., 2020), we consider the task-incremental language learning setting as that a model should learn from a sequence of tasks, where samples of former tasks cannot be accessible during the training steps for later tasks, but samples of all classes in the current task can be acquired simultaneously.

Formally, the input training stream consists of $K$ ordered tasks $\mathcal{T}_1, \mathcal{T}_2, \cdots, \mathcal{T}_K$, where we observe $n_k$ samples, denoted by $\left\{ \left( \boldsymbol{x}_i^k, y_i^k \right) \right\}_{i=1}^{n_k}$, drawn from distribution $\mathcal{P}_k(\mathcal{X}, \mathcal{Y})$ of task $\mathcal{T}_k$. Our training objective is a general model $f_\theta : \mathcal{X} \mapsto \mathcal{Y}$ which handles all tasks with a limited number of parameters $\theta$, by minimizing the negative log-likelihood averaged over all examples:

$$\mathcal{L}(\theta) = -\frac{1}{N} \sum_{i=1}^{N} \ln P\left( y_i \,|\, \boldsymbol{x}_i \,;\, \theta \right),$$

where $N = \sum_{t=1}^{K} n_t$ is the number of all training examples.

### 2.1 INVESTIGATED MODEL

In Natural Language Processing, a model can be divided into two parts, a text encoder and a task decoder, with parameters $\theta^{enc}$ and $\theta^{dec}$, respectively.

**Text Encoder**  Similar to MbPA++ (d'Autume et al., 2019) and Meta-MbPA (Wang et al., 2020b), we use BERT$_{\text{BASE}}$ (Devlin et al., 2019) as our text encoder, which produces vector representations according to given tokens.

In text classification, we take the representation of [CLS] token added at the first to aggregate information of all tokens. For a sequence of input tokens $\boldsymbol{x}_i$, where $x_{i,0}$ is [CLS], BERT$_{\text{BASE}}$ will generate corresponding vectors $\{\boldsymbol{v}_{i,j}\}_{j=1}^{L}$ with $L = |\boldsymbol{x}_i|$. Therefore, we formulate the output of encoder model as: $f_{\theta^{enc}}(\boldsymbol{x}_i) = \boldsymbol{v}_{i,0}$.

For extractive question answering, we take the task setting of SQuAD 1.1 (Rajpurkar et al., 2016), as in previous work (d'Autume et al., 2019). The input tokens $\boldsymbol{x}_i$ here are the concatenation of a context $\boldsymbol{x}_i^{\text{ctx}}$ and a query $\boldsymbol{x}_i^{\text{que}}$ separated by a special token [SEP].

**Task Decoder**  For text classification, we add a linear transformation and a soft-max layer after BERT$_{\text{BASE}}$ encoder. Following d'Autume et al. (2019), we adopt a united decoder for all classes of different tasks, and here $\theta^{dec}$ is the combination of $\{\boldsymbol{W}_y\}_{y \in \mathcal{Y}}$:

$$P(\hat{y} = \alpha | \boldsymbol{x}_i) = \frac{\exp\left(\boldsymbol{W}_\alpha^\top f_{\theta^{enc}}(\boldsymbol{x}_i)\right)}{\sum_{y \in \mathcal{Y}} \exp\left(\boldsymbol{W}_y^\top f_{\theta^{enc}}(\boldsymbol{x}_i)\right)} = \frac{\exp\left(\boldsymbol{W}_\alpha^\top \boldsymbol{v}_{i,0}\right)}{\sum_{y \in \mathcal{Y}} \exp\left(\boldsymbol{W}_y^\top \boldsymbol{v}_{i,0}\right)},$$

For question answering, the models extract a span from the original context, i.e., determining the start and end boundary of the span. Our decoder for QA has two parts of linear layers $\boldsymbol{W}_{\text{start}}$ and $\boldsymbol{W}_{\text{end}}$ for the start and the end, respectively. The probability of the $t$-th token in context as the start of the answer span can be computed as:

$$P\left(\text{start} = x_{i,t}^{\text{ctx}} | \boldsymbol{x}_i^{\text{ctx}}; \boldsymbol{x}_i^{\text{que}}\right) = \frac{\exp\left(\boldsymbol{W}_{\text{start}}^\top \boldsymbol{v}_{i,t}^{\text{ctx}}\right)}{\sum_{j=1}^{L^{\text{ctx}}} \exp\left(\boldsymbol{W}_{\text{start}}^\top \boldsymbol{v}_{i,j}^{\text{ctx}}\right)},$$

where $L^{\text{ctx}}$ is the length of context, and the probability of the end boundary has a similar form. When predicting, we consider the probability distributions of two boundaries as independent.

## 2.2 Sparse Experience Replay

In reality, humans rely on reviews to keep long-term knowledge, which is based on episodic memories storing past experiences. Inspired by this, Gradient Episodic Memory (Lopez-Paz & Ranzato, 2017) and other methods introduce a memory module $\mathcal{M}$ to the learning process. Training examples then can be stored in the memory for rehearsal at a predetermined frequency.

**Construction of Memory**  Every seen example is added to the memory by a fixed rate $\gamma$ during training. If we sample $n_k$ examples of the $k$-th task, in expectation there will be $\gamma n_k$ additional instances in $\mathcal{M}$ after learning from $\mathcal{T}_k$.

**Principles of Replay**  For experience replay, we need to set a fixed sparse replay rate $r$. Whenever the model has learned from $N_{tr}$ examples from current task, it samples $\lfloor r N_{tr} \rceil$ ones from $\mathcal{M}$ and re-learns. We set storage rate $\gamma = 0.01$ and replay frequency $r = 0.01$ in all of our experiments to ensure comparability, the same as prior work. In this paper, we name a model by **REPLAY** only if it is enhanced by sparse memory replay without other modifications. We name a model trained on a sequence of tasks without any memory replay by **SEQ**.

## 2.3 Datasets

To provide comparable evaluation, we employ the same task incremental language learning benchmark introduced by MbPA++. Its text classification part is rearranged from five datasets used by Zhang et al. (2015), consisting of 4 text classification tasks: news classification (AGNews, 4 classes), ontology prediction (DBPedia, 14 classes), sentiment analysis (Amazon and Yelp, 5 shared classes), topic classification (Yahoo, 10 classes). Following d'Autume et al. (2019) and others, we randomly choose 115,000 training and 7,600 testing examples to create a balanced collection. Since Amazon and Yelp are both sentiment analysis datasets, their labels are merged and there are 33 classes

in total. In all our experiments, we evaluate model's performance on all five tasks and report the macro-averaged accuracy as prior work.

As for question answering, this benchmark contains 3 datasets: SQuAD 1.1 (Rajpurkar et al., 2016), TriviaQA (Joshi et al., 2017), and QuAC (Choi et al., 2018). Since TriviaQA has two sections, *Web* and *Wikipedia*, considered as two different tasks, this benchmark totally consists of 4 QA tasks.

## 3 PROBING FOR INTRINSIC ABILITY AGAINST FORGETTING IN BERT

As mentioned in Section 1, a model can rapidly recover its performance of previously learned tasks, by memory replay on merely 100 instances (d'Autume et al., 2019). If the model completely loses the ability to encode prior tasks, it is counter-intuitive that the model can regain prior knowledge by 4 updating steps. We conjecture that BERT can actually retain old knowledge when learning new tasks rather than catastrophically forgetting. To verify this hypothesis, we first conduct a pilot study.

We implement our pilot experiments on the text classification benchmark, employing $\text{BERT}_{\text{BASE}}$ with a simple linear decoder as our model and training it under 4 different orders (detailed in Appendix A). Following previous probing studies (Tenney et al., 2019; Jawahar et al., 2019) to examine BERT's encoding ability for specific tasks, we freeze encoder parameters after sequentially finetuning, re-initialize five new linear probing decoders and re-train them on five tasks separately. We find that evaluated on the corresponding tasks, every fixed BERT encoder combined with its new decoder can achieve a superior performance. Surprisingly, the macro-averaged accuracy scores of all tasks for 4 orders are $75.87\%_{\pm 0.73\%}$, $76.76\%_{\pm 0.64\%}$, $75.19\%_{\pm 0.43\%}$, $76.76\%_{\pm 0.71\%}$, which are close to the performance of a multi-task learning model ($78.89\%_{\pm 0.18\%}$). However, previous works (Biesialska et al., 2020) show that sequentially trained models suffer from *catastrophic forgetting* and sacrifice their performance on previous tasks when adjusting to new task. Our pilot experiments, in contrary to previous works, actually indicate that BERT may have the ability to maintain the knowledge learned from previous tasks in a long term.

### 3.1 PROBING METHOD

To verify whether BERT can refrain from forgetting without the help of memory replay, we need a tool to systematically measure a model's encoding ability for previous tasks when it incrementally learns a sequence of tasks. One way is to compare the encoding ability of models at different learning stages trained under two different settings, REPLAY and SEQ. For each setting, we consider to measure the performance *before* learning corresponding tasks can be regarded as baselines, which indicate BERT's inherent knowledge acquired from pre-training tasks. And then we can examine to what extent BERT forgets old knowledge, by comparing the results *during* and *after* learning corresponding tasks. Therefore, it is essential to track the change of BERT's task-specific encoding ability across time. We extract parameters of the encoder and save them as checkpoints at an assigned frequency during training. In both REPLAY and SEQ, we record checkpoints every 5,000 training examples[3], without regard to the retrieval memory subset.

For every checkpoint, we probe its encoding ability for every task $\mathcal{T}_k$ by following steps:

1. Add a reinitialized probing decoder to the parameters of $\text{BERT}_{\text{BASE}}$ in this checkpoint.
2. Train the recombined model with all data in $\mathcal{T}_k$'s training set $\mathcal{D}_k^{tr}$, with $\theta^{enc}$ **fixed**, which means we adjust the parameters of probing decoder only.
3. Evaluate the scores[4] of re-trained models on the test set of $\mathcal{T}_k$.

We re-train a compatible probing decoder on a specific task without touching the encoder before evaluation. We use a linear decoder as probing network for text classification, and two linear boundary decoders for question answering, the same setting as MbPA++ (d'Autume et al., 2019) and Meta-MbPA (Wang et al., 2020b). We have to mention that there still exist some controversies on whether we should use a simpler probing decoder or a more complex one (Belinkov, 2022). Here, we adopt simple one-layer probing networks for two reasons. Firstly, a simpler probe can bring about less influence to the performance of re-trained models (Liu et al., 2019a; Hewitt & Liang,

---

[3]Since every batch has 32 instances which is not divisible by 5,000, we save parameters at the closest batches to scheduled points in order to refrain from unnecessary disturbance.

[4]We use **accuracy scores** for text classification, and **F1 scores** for extractive question answering.

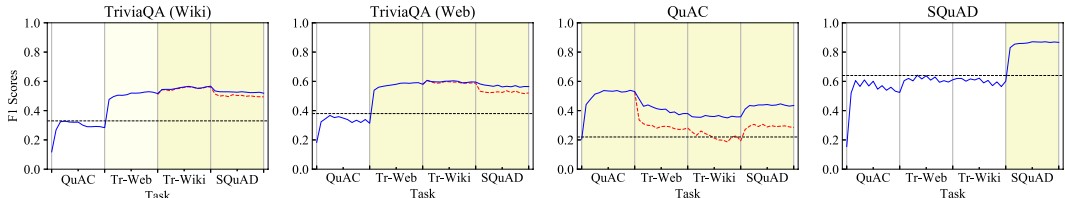

Figure 1: Probing results on five text classification tasks trained by **Order 1**, illustrated separately by the datasets[6]. The abscissas indicate the training order of tasks.

Figure 2: Probing F1 scores on four tasks trained by **Order 1**, illustrated separately[7].

2019), which enables us to focus on the encoding ability of BERT only. Secondly, our purpose in this paper is not to compare BERT's encoding ability among different tasks, but to examine whether it forgets the knowledge of a specific task. Therefore, it is better to use the same single-layer decoder as d'Autume et al. (2019) and Wang et al. (2020b), which can yield comparable results with them.

## 3.2 RETHINKING CATASTROPHIC FORGETTING

We are now able to quantitatively measure whether a BERT model can maintain its encoding ability for previous tasks during task-incremental learning, by tracking the probing scores among checkpoints. It is also important to investigate whether replay intervals have influence on BERT's encoding ability. We first set up a series of experiments on text classification described as below.

To compare with prior works (d'Autume et al., 2019; Wang et al., 2020b), we retain consistent experimental setups with them, where the maximum length of tokens and batch size are set to 128 and 32, separately. We use the training settings of REPLAY in d'Autume et al. (2019) as the baseline, which samples 100 examples from $\mathcal{M}$ for replay every 10,000 new examples from data stream. As mentioned in Section 2.2, we control storage rate $\gamma$ and replay frequency $r$ both at $1\%$. To explore the impact of memory replay, we compare models trained under different replay intervals. We randomly select a subset $\mathcal{S}$ with $\lfloor 0.01 N_{tr} \rfloor$ samples from $\mathcal{M}$ after learning every $N_{tr}$ examples. $N_{tr}$ is set to $\{10k,\ 30k,\ 60k,\ 115k\}$, and furthermore, we can consider $N_{tr}$ as $+\infty$ when training models sequentially. We employ Adam (Kingma & Ba, 2015) as the optimizer.

We use the method in Section 3.1 to evaluate the quality of the representations generated by BERT in every checkpoint. If the set of BERT parameters have a stronger ability to encode specific task, we can observe a better probing performance. Here, for text classification, we depict the changes of accuracy scores on different figures according to task and training order. The results of Order 1 (detailed in Appendix A) is shown in Figure 1 and the rest is illustrated in Appendix B. Comparing the scores before and after the model learning specific tasks, we further obtain a new understand-

---

[6]The leftmost sub-figure depicts how a model's probing accuracy scores on the training set of *AGNews* are changing along with the training procedure. The following four sub-figures are for *Amazon*, *DBPedia*, *Yahoo*, and *Yelp*. We color the background into yellow since the model is trained on corresponding task. Specially, *Amazon* and *Yelp* share the same labels, therefore, we color their background into light-yellow once the model is trained on the other task.

[7]The leftmost is *TriviaQA (Wiki)*, followed by *TriviaQA (Web)*, *QuAC*, and *SQuAD*. The F1 scores after re-training probing decoders are represented by blue lines. As a comparison, we draw F1 scores of models with original decoders by red dashed lines since the models begin to learn new tasks. We color the background into yellow since the model is trained on corresponding task. Specially, *TriviaQA (Wiki)* and *TriviaQA (Web)* are actually subsets of one task, therefore, we color their background into light-yellow when learning the other task.

ing about the task-incremental language learning: **In spite of data distribution shift among tasks, BERT remains most of the ability to classify previously seen tasks, instead of catastrophic forgetting.** This conclusion can also apply to SEQ, whose replay frequency is considered as $+\infty$. Although BERT's representation ability gets a little worse under a larger replay interval (such as 60k, 115k, $+\infty$), it still maintains previous knowledge and can recover rapidly by sparse replay.

We also provide experimental results on question answering, which is more complex than text classification. To examine whether BERT can still retain old knowledge on QA tasks, we adopt a more strict experimental setting than d'Autume et al. (2019). We train the model sequentially with 4 different orders in Appendix A, under the setting of SEQ **without any memory replay**. On each task, the model is finetuned for 15K steps, which is two times more than d'Autume et al. (2019). We then evaluate the encoding ability of every BERT checkpoints by our probing methods. The results of **Order 1** is illustrated in Figure 2, and others in Appendix C. Based on our experiment settings, the model is finetuned for enough steps to overfit on every task. However, the probing results (blue lines) are still much higher than the original scores measured before re-training decoders (red dashed lines). Comparing the obvious gap between them[8], we can find that BERT still keeps most of knowledge of previous tasks when learning new ones.

Additionally, we also investigate the ability of other pre-trained language models to retain old-task knowledge, which is detailed in Appendix D. In general, all of these pre-trained language models have an intrinsic ability to refrain from forgetting when learning a sequence of tasks, although our investigated models have various attention mechanisms and various scales. Among different training orders, they still maintain the ability to encode the first learned task, even after learning 5 tasks.

## 4 A NEW VIEW OF FORGETTING

From the experiments in Section 3.2, we observe that BERT has the potential to keep a long-term ability to provide high-quality representations for a task, once the model has learned it. Thus, it seems that we only need to finetune the decoder if we attempt to recover the model's ability for previous task. But on the other hand, the SEQ models suffer from a serious performance degradation on learned tasks, which is known as catastrophic forgetting. To reconcile this contradiction, we employ t-SNE toolkit (van der Maaten & Hinton, 2008) and visualize the representations after training on all tasks by SEQ or REPLAY (Figure 3). When learning sequentially, it shows the model produces representations of different tasks in overlapped space. In this circumstance, the task decoder identifies all vectors as instances from new task, which leads to confusion but can be averted effectively by sparse replay.

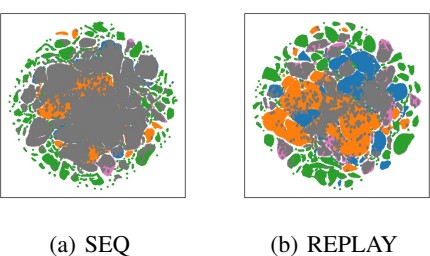

|          |            |
|:--------:|:----------:|
| (a) SEQ  | (b) REPLAY |

Figure 3: Visualization results of representation space after training on tasks by **Order 1**. Points of AGNews, Amazon & Yelp, DBPedia, Yahoo are colored by blue, orange, green, pink, respectively, while intersection areas of multiple tasks are grey.

All these observations push us to make the assumption that the forgetting in task-incremental learning can be considered as two parts, *intra-task forgetting* and *inter-task forgetting*. The *intra-task forgetting* describes whether a model can still generate meaningful representations for prior tasks after learning new ones, while the *inter-task forgetting* refers to whether the representations produced for different tasks are distinguishable from each other. In this section, we first propose a toolkit to describe the representation (in Section 4.1). Then, we exhibit the changes of a model learning continuously under REPLAY settings, and provide a novel understanding for catastrophic forgetting in NLP models. Admittedly, question answering models usually involve interactions among representations of different granularities (from token-level to even document-level) (Wang et al., 2018), thus is more challenging to analyze. Therefore, we will put more emphasis on analysing the results of text classification.

---

[8]In QA, the F1 scores on previous tasks will not decrease to zero when learning new tasks, since all QA tasks share the same answer boundary decoder. But different text classification tasks utilize different dimensions in the decoder, which leads to more drastic deterioration on scores of old tasks.

## 4.1 DEFINITION OF REPRESENTATION SUBSPACE

As claimed in Gao et al. (2019) and Wang et al. (2020a), when trained by single-layer linear decoders, pre-trained language models produce token-level embedding vectors in a narrow cone. We observe that this conclusion applies to not only token-level representations but also sentence-level representations (more details in Appendix E). Representation vectors of the same class are aggregated together, which enables us to use a convex cone to cover these vectors, whose vertex is the *origin*. To describe the vectors precisely, the cone should cover all vectors and be as narrow as possible. Formally, we denote the surrounding cone as:

$$\left\{ \boldsymbol{x} \in \mathbb{R}^d \mid \frac{\boldsymbol{x}^{\mathrm{T}} \boldsymbol{c}}{\|\boldsymbol{x}\|_2 \cdot \|\boldsymbol{c}\|_2} \geq \delta \right\} \quad (1)$$

where $\boldsymbol{c} \in \mathbb{R}^d$ is the central axis of the cone, and $\delta$ controls the filed angle.

To acquire the narrowest cone containing all vectors output by BERT, supposing the vector set is $\mathcal{V} = \{\boldsymbol{v}_i\}_{i=1}^{n}$, we solve the optimization objective described as below:

---

**Algorithm 1:** Calculating the Representation Cone

**Input:** vector set $\mathcal{V}$, input size $n = |\mathcal{V}|$, initial central axis $\boldsymbol{c}_0$, learning rate $\alpha$, termination condition $\varepsilon$
**Output:** central axis of the cone $\boldsymbol{c}$
**while** $|\mathcal{V}| > \lceil 0.95n \rceil$ **do**
    Initialize $\boldsymbol{c} = \boldsymbol{c}_0$
    **repeat**
        Compute optimization objective by Eq. 3.
        Obtain the gradient $\nabla f_\delta(\boldsymbol{c},\ \mathcal{V})$.
        $\boldsymbol{c} \leftarrow \boldsymbol{c} + \alpha \nabla f_\delta(\boldsymbol{c},\ \mathcal{V})$
        $\boldsymbol{c} \leftarrow \boldsymbol{c}/\|\boldsymbol{c}\|_2$
        Adjust $\alpha$ by linear search.
    **until** $\forall c_j$ in $\boldsymbol{c}$, $\Delta c_j < \varepsilon$
    Calculate the cosine of $\boldsymbol{v}_i$ and $\boldsymbol{c}$, denoting as $\{s_i\}_{i=1}^{|\mathcal{V}|}$. Sort $\{s_i\}_{i=1}^{|\mathcal{V}|}$.
    $m \leftarrow \lceil (|\mathcal{V}| - \lceil 0.95n \rceil)/2 \rceil$
    Select $m$ lowest $s_i$ and their relevant vectors $\mathcal{V}^{\mathrm{del}}$.
    $\mathcal{V} \leftarrow \mathcal{V} - \mathcal{V}^{\mathrm{del}}$
    $\boldsymbol{c}_0 \leftarrow \boldsymbol{c}$
**end while**

---

$$\underset{\boldsymbol{c},\ \delta}{\mathrm{minimize}}\ -\delta; \ \ \mathrm{s.t.}\ \forall \boldsymbol{v}_i \in \mathcal{V}, \frac{\boldsymbol{v}_i^{\mathrm{T}} \boldsymbol{c}}{\|\boldsymbol{v}_i\|_2} \geq \delta, \quad \|\boldsymbol{c}\|_2 = 1, \quad (2)$$

where $\|\cdot\|_2$ means L2-norm. To obtain a definite solution, we add a restriction $\|\boldsymbol{c}\|_2 = 1$, otherwise the equation implies the direction of $\boldsymbol{c}$ only without length. The representation vectors are clustered, so we can obtain a cone with a tiny file angle ($\delta \gg 0$). Therefore, Eq. (2) is a convex optimization objective, which can be solved by Sequential Least Square Programming (Kraft, 1988; Boggs & Tolle, 1995). In iteration, we acquire the optimization gradient by following expression:

$$f_\delta(\boldsymbol{c},\ \{\boldsymbol{v}_i\}_{i=1}^{n}) = \max_i \left\{ \frac{\boldsymbol{v}_i^{\mathrm{T}} \boldsymbol{c}}{\|\boldsymbol{v}_i\|_2} \right\}$$

$$\nabla f_\delta(\boldsymbol{c},\ \{\boldsymbol{v}_i\}_{i=1}^{n}) = \frac{\boldsymbol{v}}{\|\boldsymbol{v}\|_2}, \quad \boldsymbol{v} = \arg\max_{\boldsymbol{v}_i} \left\{ \frac{\boldsymbol{v}_i^{\mathrm{T}} \boldsymbol{c}}{\|\boldsymbol{v}_i\|_2} \right\} \quad (3)$$

Furthermore, to reduce the interference from outliers caused by noisy annotations, we modify the constraint conditions as that the cone only needs to cover no less than 95% training examples. Since it violates the convexity of the original objective, we employ an iterative method and get an approximate solution, which keeps every calculating step convexity-preserving. Algorithm 1 outlines the detailed solving procedure. It is obvious that cone axis should be at the center of vectors, thus we initialize $\boldsymbol{c}_0 = \sum_i \boldsymbol{v}_i / \|\sum_i \boldsymbol{v}_i\|_2$.

## 4.2 INTRA-TASK FORGETTING

From the results in Section 3.2, we find that BERT can maintain previously learned knowledge in a long term. When working with a re-trained new decoder, BERT can still perform well on prior tasks, indicating that BERT rarely suffers from *intra-task forgetting*. To investigate the mechanism preventing BERT from *intra-task forgetting*, we train a BERT model on AGNews and Amazon as an example[9] to analyse the changes within the BERT's representation space. We first train the model on all instances of AGNews, and then sample 30K instances from Amazon as the second task for task-incremental learning. Similar to Figure 1, BERT can still generate high-quality representations

---

[9]Choose by dictionary order.

for AGNews after learning Amazon without Episodic Replay. We guess after learning a new task, the representation sub-space of old tasks is still topologically ordered[10].

As shown in Figure 3(a), we can see that, without Episodic Replay, the representation vectors of old-task instances will rotate to the overlapping sub-space of the new task, which causes the decoder cannot distinguish which task the input instance should belong to. On the other hand, if we adopt a task-specific decoder (e.g., the probing decoder), it can effectively determine the class of a given instance. This may imply that the vectors of the same old-task class are still not far from each other, but they are far away to the vectors of other classes from the the same old task. Therefore, we guess if two representation vectors are trained to be at adjacent positions , they will still be neighbors after learning a new task.

To examine whether the rotating process of old-task representation vectors is topologically ordered, we first need a metric to define the relative positions among the representations of instances in the same class. Following our method in Section 4.1, we can describe the representation sub-space of a class $y$ as a convex cone, whose cone axis is $c_y$. Then, for instance $i$ of class $y$, we can define the relative position of its representation vector $v_{y,i}$ as the cosine between $v_{y,i}$ and $c_y$.

Since we need to compare the relative positions of every instance at two checkpoints (*before* and *after* learning the second task), we distinguish the vectors at different checkpoints according to their superscripts. Formally, we denote the cone axis and the representation vectors *before* learning Amazon as $c_y^{(0)}$ and $v_{y,i}^{(0)}$, with the ones *after* learning Amazon as $c_y^{(1)}$ and $v_{y,i}^{(1)}$, respectively.

For every $v_{y,i}^{(0)}$ in the $\mathcal{V}_y^{(0)}$ (the universal representation set of class $y$ before learning Amazon), we select its $n$ nearest neighbors from $\mathcal{V}_y^{(0)} - \left\{ v_{y,i}^{(0)} \right\}$ by Euclidean distance, and record their indicator set as $N_{y,i}$. It is reasonable to believe that these $n$ neighbors have the most similar semantic information to $v_{y,i}^{(0)}$. Then, we can check whether $v_{y,i}^{(1)}$ and the vectors $\left\{ v_{y,k}^{(1)} \right\}_{k \in N_{y,i}}$ are still neighbors, to verify whether the representation sub-space of class $y$ is topologically ordered. Here, we compute the the correlation between the relative positions of $v_{y,i}^{(1)}$ and $\left\{ v_{y,k}^{(1)} \right\}_{k \in N_{y,i}}$, which is estimated by Pearson correlation coefficient between $\cos(c_y^{(1)}, v_{y,i}^{(1)})$ and $\sum_{k \in N_{y,i}} \cos(c_y^{(1)}, v_{y,k}^{(1)})$. We list the results of all classes in AGNews with different scales of $n$ in Table 1 (where $y \in$ {Class-1, Class-2, Class-3, Class-4}, $n \in \{5, 10, 25, 50, 100\}$). By comparing different $n$, we can see a median size of neighbors brings a better correlation, which restrains randomness from a tiny set and uncorrelated bias from a huge set. Altogether, the influence of $n$ is inessential and we can reach the conclusion that the positions of $v_{y,i}^{(0)}$ and its neighbors are still close after learning new task, since the Pearson coefficients are no less than $0.483$ (partly higher than $0.723$).

In other words, if two examples are mapped to near positions before learning new tasks, they will remain close with each other after learning new tasks. Once BERT has learned a task, it will tend to generate representations of the same class at close positions, while generating representations of different classes at non-adjacent spaces. Therefore, if the rotating process of old-task representations can keep topologically ordered, the representation vectors of a class will always be separate to the vectors of other classes. This is why BERT exhibits an aptitude to alleviate intra-task forgetting in our study.

### 4.3 INTER-TASK FORGETTING

Neural network models always suffer from catastrophic forgetting when trained on a succession of different tasks, which is attributed to inter-task forgetting in this work. Similar to prior evaluation, we continue to use covering cones to investigate the role of memory replay when models resisting inter-task forgetting.

---

[10]Given a non-empty vector set $\mathcal{V}$, we can cluster it into many disjoint sub-sets, $\mathcal{V}^1, \cdots, \mathcal{V}^K$, by the distances between vectors. After learning a new task, the representation vectors of previous tasks will rotate to new directions. For any sub-set $\mathcal{V}^p$ and any new vector $v_x^p$ within $\mathcal{V}^p$, if any new vector $v_y^p \in \mathcal{V}^p$ is closer to $v_x^p$ than any vectors $v_z^q \in \mathcal{V}^q$ ($q \neq p$) in other sub-sets, we will think the rotating process of representation vectors is perfectly *topologically ordered* when learning new task.

Table 1: Pearson correlation coefficient ($\times 100$) of the angles of $\boldsymbol{v}_{1,i}$ and its $n$ neighbors to the cone axis. The highest scores are made **bold**, with the second underlined.

| $n$ | Class 1 | Class 2 | Class 3 | Class 4 |
|-----|---------|---------|---------|---------|
| 5 | $81.09_{\pm3.55}$ | $48.35_{\pm 9.82}$ | $83.11_{\pm3.57}$ | $72.41_{\pm3.53}$ |
| 10 | $\mathbf{81.68}_{\pm3.26}$ | $50.44_{\pm10.29}$ | $\mathbf{83.90}_{\pm3.52}$ | $\underline{73.80}_{\pm3.22}$ |
| 25 | $\underline{81.10}_{\pm3.19}$ | $\mathbf{51.46}_{\pm10.27}$ | $\underline{83.76}_{\pm3.58}$ | $\mathbf{73.98}_{\pm3.11}$ |
| 50 | $80.03_{\pm3.30}$ | $\underline{51.06}_{\pm10.56}$ | $83.25_{\pm3.65}$ | $73.39_{\pm3.12}$ |
| 100 | $78.51_{\pm3.49}$ | $50.16_{\pm10.58}$ | $83.27_{\pm3.84}$ | $72.35_{\pm3.12}$ |

When a model decodes representation vector $\boldsymbol{v}$ via a linear layer connected by soft-max, the decoder can be regarded as a set of column-vectors (i.e. $\{\boldsymbol{w}_y\}_{y \in \mathcal{Y}}$ in Section 2.1) and the predicting process is equal to selecting one having the largest inner product with $\boldsymbol{v}$. Therefore, it is necessary to check whether the cones of previous task rotate to their corresponding column-vectors in decoder. In this section, we still examine the model trained on AGNews first and continuously trained on Amazon with a replay interval of 30K for three times.

We observe that there is no significant change of column-vectors in decoder before and after memory replay, since their rotation angles are less than $1 \times 10^{-3}$, which are negligible. For each time $t$, we denote the cone axis of class $k$ before and after replay as $\boldsymbol{c}_{t,k}^-$ and $\boldsymbol{c}_{t,k}^+$, respec-

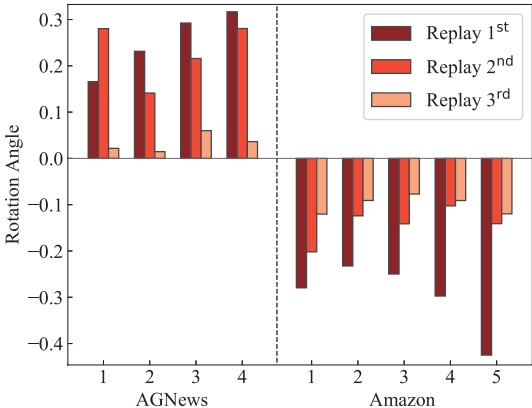

Figure 4: Bar chart for rotation angles during replay, clusters by task label and colored according to replay time.

tively, and its corresponding column-vector in decoder as $\boldsymbol{w}_k$. Then, the rotation angle of the $k$-th cone can be estimated as: $\Delta\zeta_{t,k} = \cos(\boldsymbol{c}_{t,k}^-, \boldsymbol{w}_k) - \cos(\boldsymbol{c}_{t,k}^+, \boldsymbol{w}_k)$. If $\Delta\zeta_{t,k} > 0$, it means cones rotate closer to the direction of $\boldsymbol{w}_k$ during replay. The results illustrated in Figure 4 reveal that memory replay obliges the vectors of previous tasks rotating to their corresponding column-vectors in decoder efficiently, while dragging those of current task to deviate from optimal position. Furthermore, this dual process weakens along with the increase of replay times. Since the representation space of BERT is high-dimensional while our tasks are finite, alternately learning on memory and current tasks can separate encoding vectors by mapping them to different sub-spaces.

In Appendix F, we provide more visualization results about how memory replay reduces inter-task forgetting, in other words, catastrophic forgetting in the traditional sense.

## 5 CONCLUSION

In this work, we conduct a probing study to quantitatively measure a PLM's encoding ability for previously learned tasks in a task-incremental learning scenario, and find that, different from previous studies, when learning a sequence of tasks, BERT can retain its encoding ability using knowledge learned from previous tasks in a long term, even without experience replay. We further examine the topological structures of the representation sub-spaces of different classes in each task produced by BERT during its task-incremental learning. We find that without memory replay, the representation sub-spaces of previous tasks tend to overlap with the current one, but the sub-spaces of different classes within one task are distinguishable to each other, showing topological invariance to some extent. Our findings help better understand the connections between our new discovery and previous studies about catastrophic forgetting.

Limited by the number of tasks, we have not discussed the capacity of BERT when continuously learning more tasks. As far as we know, there is no existing method yet to measure whether a model has achieved its learning capacity and cannot memorize any more knowledge. In the future, we will extend our probing method to a longer sequence or different types of tasks and explore what amount of knowledge a large pre-trained language model can maintain.

ACKNOWLEDGMENT

This work is supported by the National Key R&D Program of China (No.2020AAA0106600), and the NSFC Grants (No.62161160339).

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

## A  DATASETS AND ORDERS

For task-incremental text classification, we use the following orders to train our models, which are the same as d'Autume et al. (2019) and Wang et al. (2020b):

1. Yelp→AGNews→DBPedia→Amazon→Yahoo.
2. DBPedia→Yahoo→AGNews→Amazon→Yelp.
3. Yelp→Yahoo→Amazon→DBpedia→AGNews.
4. AGNews→Yelp→Amazon→Yahoo→DBpedia.

For task-incremental question answering, we use the following orders to train our models, which are also the same as d'Autume et al. (2019) and Wang et al. (2020b):

1. QuAC→TriviaQA (Web)→TriviaQA (Wiki)→SQuAD.
2. SQuAD→TriviaQA (Wiki)→QuAC→TriviaQA (Web).
3. TriviaQA (Web)→TriviaQA (Wiki)→SQuAD→QuAC.
4. TriviaQA (Wiki)→QuAC→TriviaQA (Web)→SQuAD.

Here, the *Web* part and the *Wikipedia* part of TriviaQA (Joshi et al., 2017) are treated as two separate datasets in the orders.

## B  PROBING ACCURACY SCORES OF ALL ORDERS FOR TEXT CLASSIFICATION

In this section, we illustrate the probing results of all four orders in Figure 5. Following the main body, background is colored by yellow when and after training on corresponding tasks. And specially, since Amazon and Yelp share the same labels, we color their background by light-yellow once the model is trained on the other.

## C  ANALYSIS FOR QUESTION ANSWERING TASKS

Similar to the analysis of text classification, we also train models on 4 question answering (QA) tasks in designated orders. To verify whether BERT has a potential to keep knowledge in a long term in QA tasks, we random sample 240K examples from each task (by repeated sampling), where their sizes are two or three times more than the original datasets. We set batch size as 16 and learning rate as $3 \times 10^{-5}$ without decay. Additionally, we do **NOT** use any memory module, which means the models are trained sequentially without memory replay.

We save checkpoints every 1,250 steps, and then re-finetune the decoders on 4 tasks respectively, with the parameters of BERT encoders fixed. Since, here, QA is formulated as a sequence-to-sequence task, there may be more than one golden answer span for a question. Therefore, we use F1 score to evaluate the performance of models. All results are illustrated in Figure 6.

The results imply $BERT_{BASE}$ still has a durability to keep previously learned knowledge in a long term in more complex tasks like question answering. In QA, the model employs unified span position decoders for all 4 tasks. therefore, the original F1 scores (before refintuning, red dashed lines) for previous tasks will not decrease to zero, which is different from text classification. Although the catastrophic forgetting problem is not too severe in QA, the models still achieve much better F1 scores after re-finetuning their decoders, considering the gaps between blue lines and red dashed lines. In the meantime, we find there is only a limited drop of blue lines after the models finish learning from corresponding tasks. It means that BERT has a satisfactory potential to keep previous knowledge, even **without any** memory replay. Our conclusions in Section 5 can also apply to question answering tasks.

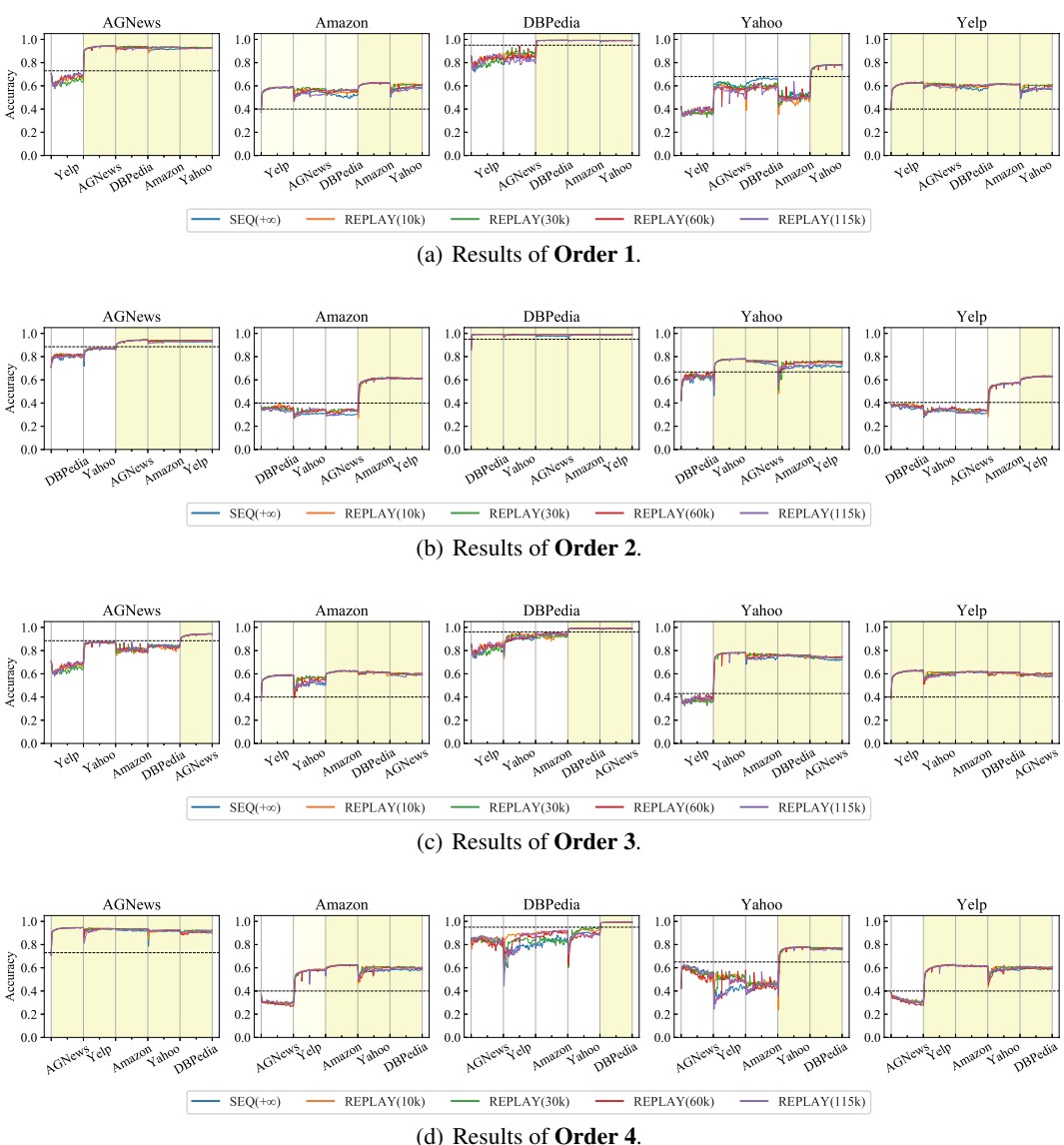

Figure 5: Probing results of five text classification tasks training by each order. In each row, we illustrate the results for 5 tasks separately, where the leftmost is AGNews, followed by Amazon, DBPedia, Yahoo, and Yelp.

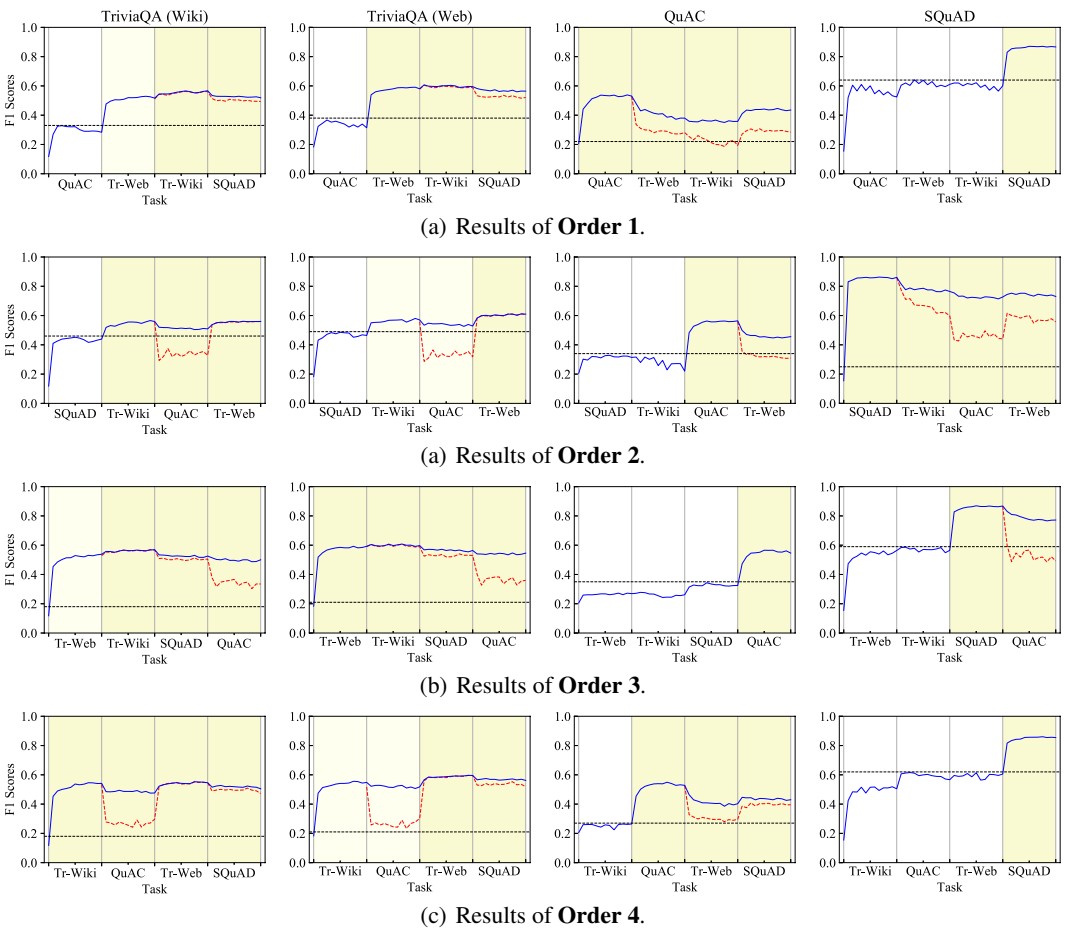

Figure 6: F1 scores on four tasks trained by 4 different orders. In each row, we plot the results for four tasks separately, where the leftmost is TriviaQA (Wiki), followed by TriviaQA (Web), QuAC, and SQuAD. The F1 scores after re-finetuning decoders is represented by blue lines, and as a comparison, we draw F1 scores before re-finetuning decoders by red dashed lines. We color the background into yellow since the model is trained on corresponding task. Specially, TriviaQA (Wiki) and TriviaQA (Web) are actually subsets of one task, therefore, we color their background into light-yellow once the model is trained on the other task.

Table 2: Probing results of various PLMs

| X | PLM | AGNews | Amazon | DBPedia | Yahoo | Yelp |
|---|---|---|---|---|---|---|
| Upper | BERT-tiny | 92.46 | 55.91 | 98.70 | 71.67 | 57.45 |
| | BERT-mini | 93.71 | 58.50 | 99.01 | 72.62 | 60.24 |
| | BERT-small | 92.01 | 54.05 | 99.09 | 73.41 | 61.17 |
| | BERT-med | 94.13 | 60.45 | 99.21 | 73.76 | 61.42 |
| | BERT-base | 94.50 | 62.41 | 99.32 | 75.08 | 62.76 |
| | BERT-large | 93.93 | 62.89 | 99.17 | 71.58 | 63.96 |
| | RoBERTa | 94.49 | 63.21 | 99.24 | 74.76 | 64.75 |
| | ELECTRA | 94.74 | 63.50 | 99.24 | 75.34 | 64.57 |
| | BART | 94.50 | 62.50 | 99.24 | 75.05 | 64.04 |
| | GPT-2 | 94.34 | 61.17 | 99.14 | 74.28 | 63.04 |
| | XLNet-base | 94.30 | 62.84 | 99.16 | 74.58 | 64.34 |
| Lower | BERT-tiny | 81.28/-11.18 | 32.00/-23.91 | 85.66/-13.04 | 49.42/-22.25 | 36.08/-21.37 |
| | BERT-mini | 82.20/-11.51 | 37.33/-21.17 | 94.43/ -4.58 | 54.70/-17.92 | 41.82/-18.42 |
| | BERT-small | 86.05/ -5.96 | 43.11/-10.95 | 97.39/ -1.70 | 61.00/-12.41 | 46.87/-14.30 |
| | BERT-med | 85.78/ -8.36 | 43.61/-16.84 | 97.47/ -1.74 | 59.74/-14.03 | 46.92/-14.50 |
| | BERT-base | 80.03/-14.47 | 42.29/-20.12 | 86.61/-12.71 | 51.17/-23.91 | 43.42/-19.34 |
| | BERT-large | 65.43/-28.50 | 35.28/-27.62 | 77.84/-21.33 | 29.20/-42.38 | 35.24/-28.72 |
| | RoBERTa | 84.54/ -9.95 | 42.25/-20.96 | 88.34/-10.89 | 56.38/-18.38 | 44.80/-19.95 |
| | ELECTRA | 72.66/-22.08 | 48.01/-15.49 | 80.63/-18.61 | 42.17/-33.17 | 49.26/-15.30 |
| | BART | 78.13/-16.37 | 43.07/-19.43 | 83.64/-15.59 | 47.68/-27.37 | 45.25/-18.79 |
| | GPT-2 | 89.55/ -4.79 | 47.70/-13.47 | 95.11/ -4.04 | 66.28/ -8.00 | 47.74/-15.30 |
| | XLNet-base | 88.50/ -5.80 | 50.75/-12.09 | 94.91/ -4.25 | 66.57/ -8.01 | 51.64/-12.70 |
| Order_1 | BERT-tiny | 87.80/ -4.66 | 40.32/-15.59 | 94.24/ -4.46 | 69.14/ -2.53 | 45.29/-12.16 |
| | BERT-mini | 88.87/ -4.84 | 42.88/-15.62 | 97.12/ -1.89 | 71.99/ -0.63 | 44.86/-15.38 |
| | BERT-small | 90.95/ -1.07 | 50.91/ -3.14 | 98.53/ -0.57 | 73.45/+0.04 | 51.92/ -9.25 |
| | BERT-med | 91.21/ -2.92 | 52.01/ -8.43 | 98.78/ -0.43 | 74.24/+0.47 | 53.13/ -8.29 |
| | BERT-base | 92.00/ -2.50 | 56.79/ -5.62 | 99.12/ -0.20 | 75.16/+0.08 | 56.43/ -6.33 |
| | BERT-large | 92.43/ -1.50 | 59.51/ -3.38 | 98.84/ -0.33 | 75.83/+4.25 | 59.01/ -4.95 |
| | RoBERTa | 92.82/ -1.67 | 60.07/ -3.14 | 98.70/ -0.54 | 75.45/+0.68 | 60.18/ -4.57 |
| | ELECTRA | 91.50/ -3.24 | 54.79/ -8.71 | 97.57/ -1.67 | 76.01/+0.67 | 55.18/ -9.38 |
| | BART | 93.66/ -0.84 | 60.82/ -1.68 | 98.78/ -0.46 | 75.64/+0.59 | 61.03/ -3.01 |
| | GPT-2 | 92.54/ -1.80 | 57.11/ -4.07 | 98.82/ -0.33 | 74.37/+0.09 | 57.36/ -5.68 |
| | XLNet-base | 92.97/ -1.33 | 61.03/ -1.82 | 98.38/ -0.78 | 75.33/+0.75 | 61.86/ -2.49 |
| Order_2 | BERT-tiny | 88.14/ -4.32 | 52.29/ -3.62 | 85.59/-13.11 | 53.72/-17.95 | 55.62/ -1.83 |
| | BERT-mini | 88.08/ -5.63 | 56.21/ -2.29 | 93.66/ -5.36 | 54.17/-18.45 | 59.18/ -1.05 |
| | BERT-small | 90.61/ -1.41 | 58.36/ +4.30 | 98.33/ -0.76 | 64.22/ -9.18 | 61.00/ -0.17 |
| | BERT-med | 91.28/ -2.86 | 59.89/ -0.55 | 98.50/ -0.71 | 64.49/ -9.28 | 61.89/+0.47 |
| | BERT-base | 91.54/ -2.96 | 61.75/ -0.66 | 99.01/ -0.30 | 64.38/-10.70 | 63.00/+0.24 |
| | BERT-large | 92.39/ -1.54 | 62.09/ -0.80 | 97.80/ -1.37 | 68.61/ -2.97 | 64.68/+0.72 |
| | RoBERTa | 93.34/ -1.14 | 63.12/ -0.09 | 98.08/ -1.16 | 69.71/ -5.05 | 64.88/+0.13 |
| | ELECTRA | 92.36/ -2.38 | 62.95/ -0.55 | 97.11/ -2.13 | 60.43/-14.91 | 65.09/+0.53 |
| | BART | 93.26/ -1.24 | 62.72/ +0.22 | 98.05/ -1.18 | 69.55/ -5.50 | 64.53/+0.49 |
| | GPT-2 | 92.71/ -1.63 | 60.88/ -0.29 | 98.42/ -0.72 | 70.51/ -3.76 | 63.61/+0.57 |
| | XLNet-base | 92.91/ -1.39 | 62.61/ -0.24 | 98.51/ -0.64 | 71.34/ -3.24 | 65.34/+1.00 |
| Order_3 | BERT-tiny | 91.39/ -1.07 | 39.46/-16.45 | 94.46/ -4.24 | 60.59/-11.08 | 45.57/-11.88 |
| | BERT-mini | 92.80/ -0.91 | 46.80/-11.70 | 96.61/ -2.41 | 64.28/ -8.34 | 48.61/-11.63 |
| | BERT-small | 93.68/ +1.67 | 54.83/ +0.78 | 98.59/ -0.50 | 68.11/ -5.30 | 55.66/ -5.51 |
| | BERT-med | 93.67/ -0.46 | 55.97/ -4.47 | 98.16/ -1.05 | 68.71/ -5.05 | 55.50/ -5.92 |
| | BERT-base | 94.28/ -0.22 | 59.09/ -3.32 | 98.66/ -0.66 | 67.46/ -7.62 | 56.97/ -5.79 |
| | BERT-large | 94.49/ +0.55 | 58.03/ -4.87 | 97.17/ -2.00 | 68.78/ -2.80 | 55.41/ -8.55 |
| | RoBERTa | 94.87/ +0.38 | 60.78/ -2.43 | 98.91/ -0.33 | 70.92/ -3.84 | 60.92/ -3.83 |
| | ELECTRA | 94.47/ -0.26 | 59.70/ -3.80 | 97.82/ -1.42 | 65.66/ -9.68 | 59.80/ -4.76 |
| | BART | 94.53/ +0.03 | 61.45/ -1.05 | 98.79/ -0.45 | 73.08/ -1.97 | 62.03/ -2.01 |
| | GPT-2 | 94.25/ -0.09 | 57.92/ -3.25 | 98.87/ -0.28 | 72.41/ -1.87 | 58.67/ -4.37 |
| | XLNet-base | 94.83/ +0.53 | 62.09/ -0.75 | 98.58/ -0.58 | 73.18/ -1.39 | 61.30/ -3.04 |
| Order_4 | BERT-tiny | 84.42/ -8.04 | 35.75/-20.16 | 98.07/ -0.63 | 66.26/ -5.41 | 42.14/-15.30 |
| | BERT-mini | 85.43/ -8.28 | 39.89/-18.61 | 98.82/ -0.20 | 68.83/ -3.79 | 44.82/-15.42 |
| | BERT-small | 88.84/ -3.17 | 50.37/ -3.68 | 99.13/ +0.04 | 70.96/ -2.45 | 53.67/ -7.50 |
| | BERT-med | 90.25/ -3.88 | 55.32/ -5.13 | 99.20/ -0.01 | 72.38/ -1.38 | 55.55/ -5.87 |
| | BERT-base | 90.91/ -3.59 | 59.62/ -2.79 | 99.33/ +0.01 | 73.78/ -1.30 | 60.17/ -2.59 |
| | BERT-large | 90.50/ -3.43 | 59.25/ -3.64 | 99.33/ +0.16 | 74.07/ +2.49 | 61.50/ -2.46 |
| | RoBERTa | 91.42/ -3.07 | 59.74/ -3.47 | 99.41/ +0.17 | 73.41/ -1.36 | 59.63/ -5.12 |
| | ELECTRA | 89.99/ -4.75 | 53.47/-10.03 | 99.24/ -0.00 | 73.12/ -2.22 | 58.91/ -5.66 |
| | BART | 92.39/ -2.11 | 60.53/ -1.97 | 99.29/ +0.05 | 75.16/ +0.11 | 60.82/ -3.22 |
| | GPT-2 | 91.84/ -2.50 | 56.11/ -5.07 | 99.16/ +0.01 | 73.39/ -0.88 | 58.54/ -4.50 |
| | XLNet-base | 92.58/ -1.72 | 61.68/ -1.16 | 99.18/ +0.03 | 74.28/ -0.30 | 62.53/ -1.82 |

## D    MORE PROBING STUDY ON OTHER PRE-TRAINED LANGUAGE MODELS

Our discussions in the main body are conducted almost exclusively on the ability of BERT$_{\text{BASE}}$ to keep knowledge. BERT (Devlin et al., 2019) is a representative of the PLM family, and widely used in various NLP tasks. We choose BERT in our study, since its transformer-based architecture influences many other PLMs. However, it does not mean BERT is the particular PLM with the intrinsic ability to generate high-quality representations for previous tasks in a long term. In this section, we further investigate various other PLMs with different model scales, different pre-training procedures, or different attention mechanisms. For the pre-trained language models with different attention mechanisms or different pre-training strategies, we investigate RoBERTa-base (Liu et al., 2019b), BART-base (Lewis et al., 2020), ELECTRA-base Clark et al. (2020), XLNet-base Yang et al. (2019), and GPT-2 (Radford et al., 2019). For the pre-trained language models with different scales, we investigate BERT-tiny, BERT-mini, BERT-small, BERT-medium, which are distilled versions from Turc et al. (2019), and BERT-large from Devlin et al. (2019). Our probing experiments are detailed as below.

To reduce redundant calculations and to provide a concise quantitative analysis, we no longer track the encoding ability of a PLM at every checkpoint. Here, we only measure the encoding ability of a PLM which has learned all tasks sequentially **without** any memory replay. All the models employ a single-layer network as decoder, which is the same as Section 3.2. And we also train models with various PLMs with identical settings to former experiments. After sequentially training on the five text classification tasks, we save the parameter weights of PLM encoder and evaluate it by probe-based method proposed in Section 3.1.

We place emphasis on that different PLMs should have different performances on a task, even if they are trained under single-task supervised paradigm. Therefore, we provide some results of *control tasks* (Hewitt & Liang, 2019) as a comparison. Specifically, we train every PLM on every dataset separately, where all parameters of encoder and decoder can be updated. These full-supervised results on single task can be consider as the **upper bounds**. To check whether a PLM itself can handle these text classification tasks well without downstream fine-tuning, we also present some zero-shot probing results as the **lower bounds**. We download the weights of various PLMs without any fine-tuning from open-source platform. Then, we train decoders for every task separately, while keeping the original PLM weights fixed (actually probing study under zero-shot scenarios). Comparing with the results of *control tasks*, we can examine whether other PLMs can retain knowledge of previous tasks like BERT, after learning a sequence of tasks.

We list all results (including the upper bounds and the lower bounds) in Table 2. From them, we can find that although these PLMs have various attention mechanisms and various scales, they have a similar intrinsic ability to keep previously learned knowledge. Although trained without Episodic Replay, these PLMs can all gain much better probing results than the lower bounds, without regard to training orders.

Comparing the results of BERT with different scales, we can find that, without Episodic Replay, the encoders with more parameters (e.g., BERT-base and BERT-large) have a little better abilities to maintain old-task knowledge than those with fewer parameters (e.g., BERT-tiny and BERT-mini). However, among the encoders with similar scales but different architectures, including BERT-base (Devlin et al., 2019), GPT-2 (base) (Radford et al., 2019), BART (Lewis et al., 2020), XLNet-base (Yang et al., 2019), they also have a similar ability to maintain old-task knowledge. Therefore, we guess this intrinsic ability to refrain from forgetting **partly comes from the scale of model**, while differences of model architectures (e.g., Transformer-Encoder v.s. Transformer-Decoder) make no obvious contributions.

## E    STRUCTURE OF REPRESENTATION SPACE

As Gao et al. (2019) and Wang et al. (2020a) mention in their work, a large pre-trained language model will embedding all words in a narrow cone when trained with a decoder like that in Section 2.1. Following their opinions, we conjecture the pre-trained language model can also generate sentence-level representation vectors of the same label in a narrow cone. To verify this consideration, we can check the cosine of any arbitrary two vectors produced by BERT. We select AGNews (Zhang et al., 2015), which has four classes, for investigation. We train a model with BERT and a linear

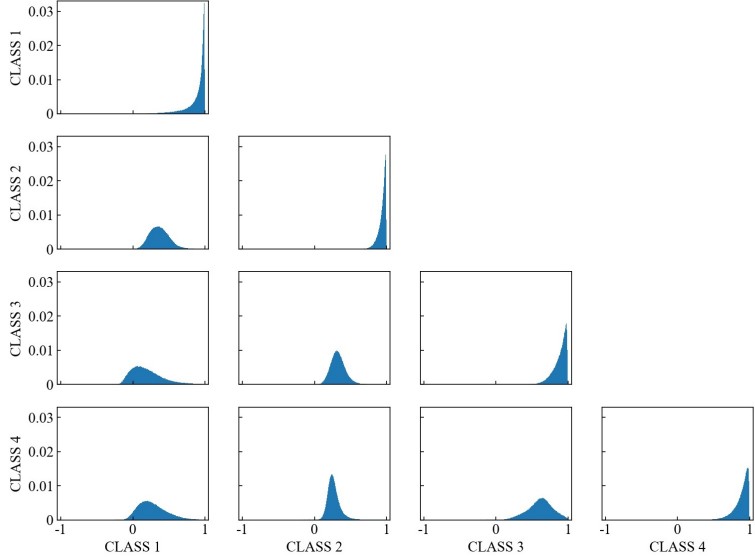

Figure 7: Cosine distribution of vectors pairs from classes of AGNews, with axes aligned.

decoder on AGNews for one pass, and then store the representation vectors of training set by class respectively. For the $i$-th and the $j$-th class ($1 \leq i \leq j \leq 4$), we randomly sample one vector from each of them for 1M times. And then, we can approximate the cosine distribution of two vectors from two classes, which illustrated in Figure 7.

From the results, it is obvious that two vectors sampled from the same class have near directions (cosine between them almost to 1), while two sample from different classes have visible discrete directions. It implies the representation sub-spaces are anisotropic, therefore, we can describe them by using convex cones.

## F ADDITIONAL VISUALIZATION RESULTS

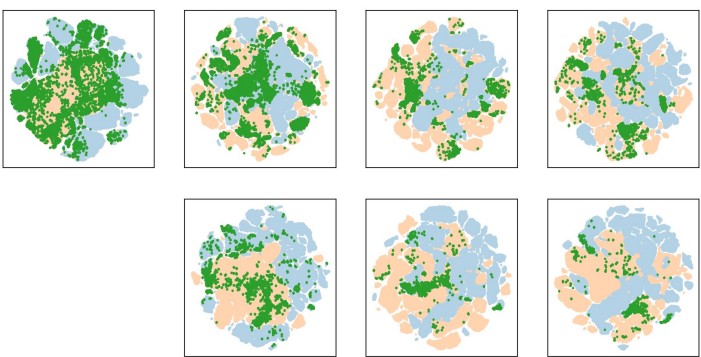

Figure 8: Additional visualization results of the representation space during lifelong learning, with points of AGNews and Amazon colored by yellow and blue respectively. Specially, we color the mixed area green, whose size should be smaller when the model has better ability to distinguish different tasks. From left to right, these columns are corresponding to the time of *just finishing learned from AGNews*, and *the first, the second, the third replay*. The top row is results before replay, while the bottom is after replay.

In this section, we visualize the change of representation space before and after memory replay during lifelong learning. Following the experiment setting in Section 4, we first train the model on AGNews, and then on Amazon with replaying three times. We save all representation vectors after

learning AGNews, and every time before or after replay. Then we adopt t-SNE (van der Maaten & Hinton, 2008) to draw all vectors in the plane. Concerning the mixed areas of both classes, we can conclude that memory replay plays a significant role to mitigate inter-task forgetting. Every time after replay, the model have a stronger ability to distinguish instances from different tasks, which is characterized by decrease of green area in Figure 8. Also, comparing the results among columns, we can confirm although it brings a little confusion among tasks when learning one task continuously without break, sparse memory replay can eliminate the confusion effectively. Therefore, a BERT model enhanced by memory replay can resist not only intra-task but also inter-task forgetting.

