# OpenReview forum: "Can BERT Refrain from Forgetting on Sequential Tasks? A Probing Study"
_ICLR.cc/2023/Conference — ICLR 2023 poster_

### Official Review · Reviewer_Lykc · 2022-10-25

**Confidence:** 3
**Correctness:** 3
**Technical Novelty And Significance:** 3
**Empirical Novelty And Significance:** 3
**Recommendation:** 6

**Clarity, Quality, Novelty And Reproducibility:**

The paper is overall well structured and presented, though as noted above I feel that the approach and justification for some parts of $\S 4$ is unclear. The quality is good and could be improved by some further contextualization of the results. The work seems substantially novel, particularly in terms of the methods and results presented in $\S 4$.

It does not seem that code or data was included in the submission, which raises questions for reproducibility, especially given that the paper proposes some novel analyses. Despite the relatively clear presentation, it might be difficult for a motivated reader to fully reproduce these method or results. I would encourage the authors to consider sharing their code.

**Strength And Weaknesses:**

Strengths:
- The overall presentation and analysis are of good quality and rest on a sequence of interesting and informative experiments. These are unified under the general theme of understanding how the representations in the BERT encoding layer evolve during training.
- The probing study is simple and effective in showing that "forgetting" is not a matter of representational degradation in the encoding layer, as decoder "probes" achieve similar accuracies both during and after training on a given task. This raises subsequent questions that the authors answer with a novel perspective.
- The authors expand on previous observations in the literature to argue that same-class sentence-level embeddings lie within a cone, then use this representation to propose and execute novel analyses relevant for the topic of task forgetting and replay. These methods represent potentially new and useful ways to study the evolution of learned representations over the course of training. The results offer new insights as to how learned representations change (or don't change) when a new task is learned or after replay.

Weaknesses:
- There is some lack of clarity in the presentation of the novel methods, particularly in $\S 4.2$. The authors should define precisely what it means for the "rotating process" to be "topological(ly) ordered". It is also unclear what it means to evaluate the "correlation between the relative positions of $v_{y,i}^{(1)}$ and $(v_{y,k}^{(1)}), k \in N_{y,i}$" or why this is "estimated by the Pearson correlation coefficient between $\cos(c_{i}^{(1)}, v_{y,i}^{(1)})$ and $\sum_{k \in N_{y,i}} \cos(c_{i}^{(1)}, v_{y,k}^{(1)})$" (in the paper the quantity $v_{y,i}^{(1)}$ appears inside the sum but I assume this is a typo). Overall, it is somewhat unclear why this specific approach was taken if the goal is just to establish that same-class embedding vectors have the same or similar nearest neighbors after training on some subsequent task.
- The authors spend relatively little time contextualizing the sequence of (interesting) results that they generate. How do these support or contradict existing hypotheses in the literature regarding the mechanism of task forgetting and replay? What hypotheses are the studies in $\S 4$ designed to evaluate, and what are the directions for future analysis?


**Summary Of The Paper:**

The authors conduct a two-part study to investigate phenomenon of "forgetting", i.e. degraded prediction performance on past data, when a language model is trained on a sequence of tasks. First, they show evidence via a probing study that a BERT encoder base in fact largely retains representations capable of high performance on a given task, even after subsequent training on other tasks. Second, they propose a sequence of analyses based on the idea that same-class embedding vectors fall in a relatively narrow cone, showing that (1) embedding vectors from the same class for a given task have similar nearest-neighbors after training on another task, and (2) replay re-aligns cone axes with the relevant decoder column for past tasks while inducing some misalignment for the current task.

**Summary Of The Review:**

This paper presents a coherent, creative sequence of empirical analyses that illuminate how the phenomena of sequential task forgetting and recovery via replay are related to the representational capacity and within-class structure of embedding vectors from a BERT encoder. The authors could further improve on their contributions by clarifying some details of a novel analysis in $\S 4.2$ and releasing their code.

---

> ### Author Response · Authors · 2022-11-13
> **Response to Reviewer Lykc**
>
> Thanks for your very helpful comments and great suggestions, which are important to improve our work. Next, we will clarify each concern with our best efforts.
>
> **For the first question:** In Section 4.2, we explain why the BERT encoder can maintain previous knowledge in a long term, as shown by the probing results. We first train a model on AGNews, and subsequently train the same model on Amazon without replay, and examine how the representing spaces of classes in AGNews change before and after learning Amazon.
>
> * **Rotation process:** For a given instance $x$ of AGNews, its representation vector before learning Amazon can be denoted as $v_x^{(0)}$. And after learning Amazon, we denote the new representation vector of $x$ as $v_x^{(1)}$. Then, we can say, after learning Amazon, representation vector of $x$ rotates from $v_x^{(0)}$ to $v_x^{(1)}$.
>
> * **Topologically ordered:** Assuming there is a non-empty vector set. We can cluster this set into many disjoint sub-sets by the distances between vectors. We consider after learning a new task, these vectors will rotate to new positions. At that time, for any new vector, if all new vectors within the same sub-set to it are closer  than the vectors out of this sub-set, we will consider the rotation process is *perfectly topologically ordered* during learning a new task.
>
> * Why can we examine whether the rotation process is topologically ordered by the Pearson correlation coefficient between the two cosines? And why we employ this specific approach?
>
>   Thank you for pointing out the typo in $ \sum_{k\in N_{y,i}}\cos(\boldsymbol{c}_{y}^{(1)},\boldsymbol{v}^{(1)}_{y,k}) $.  We think this question can be considered as two parts: **(1)** why we use cosine between a representation vector and its cone axis, but not the projection distance from the vector to its cone axis; **(2)** if two representation vectors are neighbours before learning new task, why a high Pearson correlation coefficient between their new positions after learning new task can indicate why BERT does not suffer from intra-task forgetting.
>
>   **(1)** Since we adopt a linear matrix as decoder, the decoding process can be considered as to select a maximum inner-product between a given representation vector and the column-vectors of the decoder matrix. Therefore, when describing the "position" of a representation vector, it may not be appropriate to use the projection distance, while it is better to use cosine that measures the angle between two vectors.
>
>   **(2)** We consider the model has perfectly learnt the first task before training on new tasks. And a well-trained model should provide representation vectors with a close distance for two instances with similar semantic meanings. Intuitively, we can assume that two instances of the same class should have more similar semantic meanings than two of different classes. A high Pearson score can show that, if two vectors are neighbours before learning new tasks, they will also be neighbours with a high probability after learning new tasks. That is, if the representing sub-spaces of two different classes are separable before learning new tasks, they will keep separable to each other after learning new tasks. The high Pearson scores in Table 1 reveals that, BERT can refrain from intra-task forgetting, by keeping the representation vectors of the same class converging together, but the vectors of different classes separating to each other.
>
> **For the second question:** In this work, we do NOT aim to support or contradict existing hypotheses about catastrophic forgetting. What we really want is to  provide a new perspective to interpret the mechanism of forgetting. Our probing study reveals that, without replay, BERT can still provide distinguishable representations for previous tasks in a long term, showing that BERT can refrain from intra-task forgetting. However, we also admit, without replay, BERT will lose the ability to confirm which previous task a given  instance should belong to, that is, BERT may suffer from inter-task forgetting. We argue that inter-task forgetting can be considered as the main reason for catastrophic forgetting. We also find that the popular LL strategy, Episodic Replay, actually alleviates the inter-task forgetting.
>
> We think this work can show our community pre-trained encoders (like BERT) have the potentials that are not explored before. We hope it can inspire people in future to consider the encoder and the decoder as two separate parts when designing new lifelong learning algorithms. And we expect there will be more works to take advantage of pre-trained encoders' intrinsic ability to avoid intra-task forgetting.
>
> In the future, we will investigate whether BERT can still maintain old-task knowledge when learning on a sequence of tasks with different forms, e.g, text classification, question answering, semantic parsing, etc.

---

### Official Review · Reviewer_t1W5 · 2022-10-25

**Confidence:** 2
**Correctness:** 3
**Technical Novelty And Significance:** 3
**Empirical Novelty And Significance:** 3
**Recommendation:** 8

**Clarity, Quality, Novelty And Reproducibility:**

Clarity: While a lot of the methods were easy to follow, the conclusions or observations made form each experiment were a bit difficult to parse!
Quality: Interesting paper although it could benefit from analyses suggested above.
Originality: To the best of my knowledge, this seems like a novel step towards understanding replay mechanisms for multi-task learning.

**Strength And Weaknesses:**

I appreciate the authors' detailed response. I've read through the comments and have updated my score accordingly.

##################################

Strengths:
The paper presents an interesting problem and novel use of the solutions prescribed to analyze it.

Weaknesses/Questions:
- On one hand the authors say that the task decoders are united in that the denominator for $P(\hat{y}=\alpha|x_i)$ is over all $y \in Y$. But for question answering it seems that the decoders for start and end are disjoint, and the softmax is over all tokens in the context not all tasks. Could the authors clarify? This also plays into the fact that catastrophic forgetting is evaluated by training new probes on the frozen encoder. If the decoder is separate for each class, could we not use that directly? One problem would be that perhaps the model has changed how it stores information after seeing a new task. This could be verified by testing if the original decoders work as well, in comparison to the newly trained decoders.
- Section 3 introduction: Please add standard errors here and everywhere. Just to be clear, the macro-averaged accuracy scores is based on SEQ and not REPLAY right?
- Figure 1: Nit: Put the replay order backward! Also indicate what the dashed lines mean.
- Figure 2: Where is the dashed red line for SQuaD? Nit: Order subpanels based on order of training tasks.
- Section 3.2: “However, the probing results (blue lines) are still much higher than the original scores measured before re-training de- coders (red dashed lines). Comparing the obvious gap between them4, we can find that BERT still keeps most of knowledge of previous tasks when learning new ones.” Yes but the information might be re-organized which is why the blue line does a bit better. (as the authors also note in the 4th footnote)
- Table 1 analysis: Just to be clear, the representation vectors for a particular dataset, say Amazon, is when one does inference on the Amazon training instances before training on it correct? This is done to define the cone axis and find narrowest angle? And the process is repeated after training on Amazon? I understand the method and motivation but this doesn’t test for inter-class separability no? Even if the topological order within a class might be high, the cone axes or the cones itself could be overlapping leading to reduced separability. This brings me to the point that we have no way of ascertaining that a given correlation is high or low without looking at what the value would be for two pairs sampled randomly from different classes. This would tell us about the ordering of the space entirely. Also, what are the variances for each entry in table 1? How much does this value vary for different examples?
    - I would also be curious to see how this ordering varies for different subsets of classes. Do some classes, for example, get closer/farther depending on the distribution of the most recent training set?
    - It would also be more interesting to see how this metric varies for the AGNews dataset before training, after training and then after incremental training.
- What is the expected result for the inter-task forgetting experiment and the interpretation?  If the value is positive, it means that the new angle is smaller than the old. So as you replay more, the new and old angles become almost equal and since the old angle is fixed, equality would mean that the new angle became bigger? On the contrary, a negative value means that the old angle is smaller than the new. So as you replay more, the difference between them becomes small and since the old angle is fixed, equality would mean that the new angle became smaller? If this is correct, I understand “reveal that memory replay obliges the vectors of previous tasks rotating to their corresponding column-vectors in decoder efficiently, while dragging those of current task to deviate from optimal position.” but it wasn’t clearly written in the text.
- I am confused by the contradiction between statements and results. For example, the first result (Fig. 1) suggests that the sequential task scheme does as well as memory replay but the authors go on to suggest it doesn’t for the rest of manuscript and do subsequent analyses based on this assumption. For example, the t-SNE plot suggests that under SEQ and not REPLAY, there is a high degree of inter-class overlap. How can we explain figure 1 results then? Additionally, aren’t the results in table 1 for SEQ? If this is the case, what does the high correlation tell us about purported catastrophical forgetting in SEQ? This is not clear from section 4.2 but section 4.3 leads me to believe it is replay. Please clarify.

**Summary Of The Paper:**

This work analyzes catastrophic forgetting during different multi-task learning strategies, specifically sequence training and replay. The authors test BERT on a suite of classification tasks and question answering, and use linear probes to test the ability of the model on each task before training, after training and after replay. They find that the sequential model does as well as the replay models however, t-SNE plots reveal that the sequential model often confuses representations across tasks. Next, they look at the angle between an exemplar and the mean of the class to characterize how the topological ordering changes due to replay. Finally, they look at the effect of replay in both the current dataset and the one being replayed.

**Summary Of The Review:**

Overall, the paper presents an interesting set of tools to analyze the problem of catastrophic forgetting in neural LMs. However, the results and claims seem contradictory and it is not clear to me if the authors are claiming that purely sequential learning *suffers* from forgetting or it does not! To that end, I believe that these tools can be used to more systematically understand how the replay and sequence strategies differ in the geometry of the representations given that they both do well for the tasks described here.

---

> ### Author Response · Authors · 2022-11-13
> **[Part 1] Response to Reviewer t1W5**
>
> Thanks for your helpful and constructive comments. The main finding of this work is that, without replay, a BERT encoder still has a *potential* to maintain knowledge of previous tasks during task-incrementally learning. This phenomenon seems contradictory to previous studies about catastrophic forgetting (CF). We further reveal that a BERT encoder trained without replay loses the ability to distinguish which task a given instance should belong to, but it can still produce separable representations for different classes of the same previous task, which, on the other hand, indicates that the main role of Episodic Replay is to help BERT enhance its ability to distinguish instances from different tasks.
>
> Next, we will try our best to clarify each concern with details:
>
>  * For Q1: About the decoders.
>
>     The form of decoders in our work is inherited from previous efforts in lifelong learning (LL) ([d’Autumn19; Wang20]). In the standard task-incremental learning paradigm, models should not access any task descriptor, i.e., models cannot foresee which task the input instance belongs to during inference. Therefore, in text classification, the decoder is a N-dim combined matrix, where N is the total number of classes of all tasks. However, question answering (QA) is a span-detection task, in which the answers are sub-strings of given texts (within datasets in our work, there is only one answer span for a given pair of text and question). Therefore, for each token of a given instance, QA models need to predict its probability to be start of answer span and its probability to be the end separately. For instances of any task, what QA models need to do is the same. So, the decoders for start and end should be disjoint, and all tasks share the same two decoders. It is also why softmax is over all tokens (to provide probabilities for each token), but not all tasks.
>
>     If we test directly with original decoders without Replay, the results of previous tasks will deteriorate seriously, e.g., in text classification, it can be down to zero, mentioned as catastrophic forgetting in the literature. That is why we adopt probing study to focus only on the knowledge in encoders.
>
>  * For Q2: About the macro-averaged accuracy scores in Section 3.
>
>     In Section 3 Introduction, the macro-averaged accuracy scores is based on SEQ (but evaluated after re-training new decoders with the encoders frozen).
>
>  * For Q3: What the dashed lines mean in Figure 1.
>
>     Dashed lines in Figure 1 show the best probing scores that the encoder can obtain on a specific task before learning it, which can be seen as how well the encoder can perform before learning the knowledge of this task. We can find that once the model has learned a specific task (yellow background in Figure 1), the probing scores will never be down to below dashed lines. It means the encoder can still maintain previous knowledge along with learning new tasks, under a more sparse replay, or even without replay.
>
>  * For Q4: Where is the dashed red line for SQuAD?
>
>     Red dashed lines in Figure 2 are the F1 scores evaluated with original decoder, which show the whole models (including encoder and decoder) suffer from forgetting when learning new tasks.
>
>     Since SQuAD is the last task in Figure 2, there will be no training process of new tasks after it.
>
>  * For Q5: About the unclear description in Section 3.2.
>
>     Thank you for pointing out these unclear parts. We will clarify with more details and explanations.

---

> > ### Author Response · Authors · 2022-11-13
> > **[Part 2] Response to Reviewer t1W5**
> >
> > * For Q6: **(1)** To clarify what is the representation vectors for a particular dataset. **(2)** A given correlation within the same class is high cannot ensure there is no inter-class overlapping. **(3)** Will the results vary if we use the metrics in Section 4.2 on different datasets? **(4)** What are the variances for each entry in table 1? **(5)** If a model learns more than two tasks, whether can the representing spaces of the first task still keep topologically ordered? **(6)** Do there exist relations between the changes of representing space of a previous task, and data distribution of the most recent training set?
> >
> >   **(1)** Sorry for the confusion. As mentioned in Section 2.1, the representation vectors for a particular dataset, say AGNews, are the vectors produced by BERT for instances of AGNews training-set. To be clear, what we want to examine is the changes of representation vectors of AGNews, before and after we train the encoder on Amazon.
> >
> >   **(2)** Regarding that the inter-class separability can not be examined by Pearson scores on pairs sampled from the same class, we examine what the Pearson scores will be if we select two pairs randomly from **different classes** in AGNews. For each instance of a specific class in AGNews, we randomly sample a vector pair from the other classes in AGNews. Then we calculate the Pearson scores for the four classes in AGNews separately. The results are: 0.0056, -0.0020, -0.0140, -0.0005, which show there is no overlapping between different classes of the same task.
> >
> >   **(3)** Additionally, we also use the toolkit to examine DBPedia (14 classes) and Yahoo (10 classes). Similar to the experiment trained on AGNew and Amazon, we now first train a model on DBPedia and then train it on Yahoo, and then extract the representation vectors for every training instance in DBPedia from two checkpoints: (i) when just finishing learning DBPedia and (ii) after learning Yahoo without replay. We calculate the Pearson scores for each class in DBPedia, as shown below. We can see that our conclusion in Section 4.2 can be general to more datasets than AGNews & Amazon:
> >   | | | | | | |
> >   |----|----|----|----|----|----|
> >   |Class 1-5 | 0.7296 | 0.7934 | 0.8433 | 0.8696 | 0.8956|
> >   |Class 6-10 | 0.8391 | 0.8428 | 0.8815 | 0.8688 | 0.8901|
> >   |Class 11-14 | 0.7973 | 0.9146 | 0.8717 | 0.7220|  |
> >
> >   **(4)** We do apologize that we run experiments only one time for the results in Table 1, since it costs a lot of time to obtain the narrowest cone of 28.75K instances of every class in AGNews training-set. We repeat the experiments on AGNews and Amazon for three times, and then obtain Pearson scores with the same method in Section 4.2. The revised averages and variances on three runs are listed as below:
> >   | n | Class 1 | Class 2 | Class3 | Class 4 |
> >   |----|----|----|----|----|
> >   |5 | 0.8109$\pm$0.0355 | 0.4835$\pm$0.0982 | 0.8311$\pm$0.0357 | 0.7241$\pm$0.0353|
> >   |10 | **0.8168**$\pm$0.0326 | 0.5044$\pm$0.1029 | **0.8390**$\pm$0.0352 | 0.7380$\pm$0.0322|
> >   |25 | 0.8110$\pm$0.0319 | **0.5146**$\pm$0.1027 | 0.8376$\pm$0.0358 | **0.7398**$\pm$0.0311|
> >   |50 | 0.8003$\pm$0.0330 | 0.5106$\pm$0.1056 | 0.8325$\pm$0.0365 | 0.7339$\pm$0.0312|
> >   |100 | 0.7851$\pm$0.0349 | 0.5016$\pm$0.1058 | 0.8427$\pm$0.0384 | 0.7235$\pm$0.0312|
> >
> >   **(5)** We also examine whether the represention sub-spaces can keep topological ordered in a longer term. We train a BERT model on five tasks one by one without replay: AGNews, Yelp, Amazon, Yahoo, DBPedia (Order 4 in Appendix A). Then, we compare the changes of representation vectors of every instance in AGNews at two checkpoint: (1) just finishing learning on AGNews; (2) after learning on DBPedia (the last task). We use the same method as Section 4.2. The Pearson scores for each class in AGNews are: 0.8479, 0.6935, 0.6956, 0.7281, showing that BERT has a strong ability to maintain old-task knowledge by keeping the representing spaces of previous tasks topologically-ordered.
> >
> >   We hope these results can answer the concerns about the results in Table 1. We will add these results and analysis in the revised version.
> >
> >   **(6)** [To be continued]

---

> > > ### Author Response · Authors · 2022-11-13
> > > **[Part 3] Response to Reviewer t1W5**
> > >
> > > * Q6: **(6)** Do there exist relations between the changes of representing space of a previous task, and data distribution of the most recent training set?
> > >
> > >   **(6)** Regarding the question: **how this phenomenon varies for different subsets of classes. Do some classes, for example, get closer/farther depending on the distribution of the most recent training set?** We think it is an inspirational question. We train a model first on AGNews (news classification, with four labels: *World, Sport, Business, Sci/Tech*) and then on Yahoo (web-query classification,  with ten labels: *Science \& Mathematics, Sports, Business \& Finance*, etc).  We find in AGNews, the instances of *Business* and *World* share more tokens in their vocabularies than instances of *Business* and *Sport*. Now, we can consider the instances of *Business* is more similar to *World* than *Sport*. We train the model several times with different random seeds. And we find that, after training on Yahoo, the vectors of *World* and *Business* are not always closer to each other, and the vectors of *Business* and *Sport* are  not always farther either. Additionally, we find it is not the case that the cone of *Business* of AGNews will overlap to the cone of *Business \& Finance* in Yahoo, after training on Yahoo without replay. Unfortunately, we do not find reliable patterns to interpret whether the vectors of two classes become closer or farther after learning a new task. But we definitely agree that this is a very interesting and valuable question, and we hope to investigate it in the future.
> > >
> > > * For Q7: To clarify the experiments in Section 4.3.
> > >
> > >   Sorry for the confusion. Yes, your understanding about the experiments in Section 4.3 is correct. We will clarify this in the revised version. In Section 4.3, our main purpose is to interpret why **sparse** Episodic Replay can enhance encoders to map the instances of different tasks to separate representing sub-spaces. As a comparison, without replay, the representing sub-spaces of different tasks will overlap to each other. If a score in Figure 4 is positive, it is correct that *the new angle is smaller than the old*. In text classification tasks, our decoder is a linear matrix, so the decoding process can be considered as to select the maximum inner-product between a given representation vector and all column-vectors in decoder matrix. Therefore, a positive score in Figure 4 means the representation vectors are rotating closer to the correct column-vectors. Then these instances will be correctly classified with a larger probability.
> > >
> > >   We argue that Episodic Replay can alleviate inter-task forgetting by *obliging the vectors of previous tasks rotating to their corresponding column-vectors in decoder*. And if we replay more times, the differences of angles before and after replay will be smaller, converging to zero, which shows the representing sub-spaces of different tasks have been effectively separated.
> > >
> > > * For Q8: About the contradiction between statements and results.
> > >
> > >   Does a model without Replay suffer from CF? Yes, but this conclusion is based on the viewpoint to consider the encoder and decoder as a whole model. Our work reveals that, when we only focus on the encoder’s potential to perform well on every previous task, the encoder does NOT actually lose the knowledge learnt from previous tasks, even without replay. But we have to admit that, without replay, although the encoder can provide distinguishable representations for every previous task **separately**, it cannot perform well on all previous tasks **together**.
> > >
> > >   To be clear, the results in Figure 1 are evaluated after we re-train five new decoders for five tasks, respectively. But if we re-train a new united decoders for all tasks together, the results of SEQ will be worse than REPLAY, due to the inter-task forgetting. It is also correct that results in Table 1 are for SEQ. In Section 4.2, we want to interpret why BERT has a natural ability to refrain from intra-task forgetting, therefore, we should train the model without replay.
> > >
> > >   In this work, we do not aim to design a new lifelong learning algorithm or obtain SOTA scores on several benchmarks. What we want is to show our community that pre-trained encoders (i.e., BERT) have the potentials that are not explored before. We hope this work can inspire people to consider the pre-trained encoder and the decoder as two separate parts when designing new LL algorithms.
> > >
> > > At last, we do appreciate Reviewer t1W5 to the valuable comments and suggestions for our work, which will help us to further improve our work.

---

### Official Review · Reviewer_zUuP · 2022-10-27

**Confidence:** 4
**Correctness:** 4
**Technical Novelty And Significance:** 1
**Empirical Novelty And Significance:** 1
**Recommendation:** 8

**Clarity, Quality, Novelty And Reproducibility:**

The paper is clear and easy to understand but Section 4.2 could use another pass (a strategy that works for me is to read every sentence out loud and ask what does it mean?). The novelty is marginal at best. I'd recommend the authors put a section in Appendix about implementation details and add standard errors to the results.

**Strength And Weaknesses:**

I like this paper, the results are neat, the exposition is clear (Section 4.2 could use some work as described later), and (almost) everything is easy to understand. But as I was reading this paper, I kept coming back to one paper [1] that I read last year which had done everything this paper is offering. They answered the same questions, they did an even more extensive set of experiments, and the only difference was in how they viewed why pretrained models maintain good representations even if the decoder needs to be retrained. I spent the last day reading both papers side by side and in my view, up until Section 4, there's hardly any difference between the two works. It appears that the authors may not be aware of [1] as it hasn't been cited in this paper so I invite the authors to read [1] and share if they disagree with my assessment. But in light of this, so far, I see the marginal value of this paper lying in Section 4 and would recommend the authors to amplify that as the core contribution of the paper (which is currently listed as their third contribution in Introduction), as the rest of their contributions (claimed contributions 1 and 2) have already been offered to the community previously. Additionally, please add standard errors in every result.

[1] Sanket Vaibhav Mehta, Darshan Patil, Sarath Chandar, and Emma Strubell. "An empirical investigation of the role of pre-training in lifelong learning." arXiv preprint arXiv:2112.09153 (2021).

Errata:
- Introduction, first paragraph: "learning the xxx one". No idea what you meant to say here.
- Introduction, first paragraph: "learning a sequecne of tasks" ---> "learning a sequence of tasks"
- Section 3.2, second paragraph: Kingma and Ba (2015) should be \citep not \citet
- Section 4.1 "decoders,pre-trained" ---> "decoders, pre-trained"

**Summary Of The Paper:**

In this paper, the authors investigate whether or not pretrained language models like BERT have the ability to maintain previously learned knowledge in the long term. To do this, they track the encoding ability of BERT for specific tasks before, during, and after learning new tasks. They find that BERT can actually refrain from forgetting when learning a sequence of tasks, contrary to existing studies about catastrophic forgetting. The authors believe this is due to the fact that BERT has a strong potential to produce high-quality representations for previous tasks even without memory replay. They further investigate the topological structure of the learned representation sub-space within and among different tasks and find that forgetting can be interpreted as intra-task forgetting (forgetting what has been learned within a task) and inter-task forgetting (forgetting what has been learned across tasks).

**Summary Of The Review:**

I like the paper and found the results to be clear and easy to understand. However, my current view is that much of what was presented in this paper had already been answered in another paper last year which the authors may be unaware of (as it hasn't been cited in this work). The main difference between the two papers, in the my opinion, is in Section 4. I recommend that the authors emphasize Section 4 as the main contribution of the paper.

---

> ### Author Response · Authors · 2022-11-10
> **[Part 1] Response to Reviewer zUuP**
>
> Thank you for the helpful comments and bringing us the inspirational work by Mehta et al (2021). We have to admit we did not come across this work before, otherwise we would adopt the loss contour analysis in [Mehta21] to investigate how many tasks a BERT can memorize (whether BERT can memorize infinite tasks). It seems that both [Mehta21] and ourselves start to ask the same question, whether large-scale pre-trained models naturally have an ability to alleviate forgetting. However, the perspectives to think about the ability to reduce forgetting between their work and ours are actually different, thus leads to different analysis viewpoints (with/without pre-training v.s. with/without LL algorithms) and different explanations regarding Catastrophic Forgetting (CF) (empirical results showing what can bring a smaller forgetting rate v.s. analysis on the learned representing space to reveal reasons of forgetting).
>
> As a brief summary, [Mehta21] argues that the pre-training process plays an important role to alleviate forgetting of the whole model (including the pre-trained encoder and the classifier/decoder). But our work focuses on whether the crucial part, the pre-trained encoder (i.e., BERT) forgets knowledge learned from previous tasks. By comparing the scores of models with/without pre-training, [Mehta21] finally draws a conclusion that, under the same LL strategy, a model enhanced by pre-training have a better ability to reduce forgetting. However, our work comes from how Episodic Replay enhances the model’s ability to keep providing high-quality representations for previous tasks. We discover that even without Episodic Replay and other LL strategies, a BERT encoder still has a strong potential to perform well on previous tasks.
>
> Next, we will detail the differences between our work and [Mehta21] from the following aspects.
>
> 1.  Different motivations.
>
>     [Mehta21] argues that the pre-training process can help the model to alleviate forgetting knowledge of previous tasks during learning new ones. They directly compare the accuracies over all tasks and the forgetting rates of pre-trained and randomly-initialized models, and find that among popular LL strategies of FineTune, EWC, Epsodic Replay, pre-trained models have smaller forgetting rates and better accuracy scores than randomly-initialized ones.
>
>     However, our work comes from an observation on previous efforts in [d’Autumn19]. Under the Episodic Replay strategy, when we make much more sparse reply, e.g., 10 times more sparse than [d’Autumn19] (1150 previous instances after every 1150K new-task instances compared to 100/10K), the model’s performance does not hurt much. This makes us curious regarding the impact of replay frequency in LL. We, therefore, wonder whether the model can still perform well on previous tasks, when the replay interval goes to $+\infty$, equivalent to sequentially learning without any LL strategy.
>
> 2. Different investigation perspectives
>
>     When examining how the pre-trained models reduce forgetting, [Mehta21] treats the encoding component and the classification (decoding) part as a whole. But we mainly focus on the how much previous knowledge the encoder part maintains, as prior work of model interpretability does ([Tenny19]). We wonder if the model is trained without Episodic Replay and other LL strategies, whether the BERT encoder will still provide high-quality representations for previous tasks along continuously learning. Therefore, we freeze the encoder and re-train a new decoder, which allows us to separately measure the encoder’s or the decoder’s ability to refrain from forgetting, to reduce possible interference to each other.
>
>     Compared with [d’Autumn19] and other works on ”5-dataset-NLP”, [Mehta21] uses a smaller learning rate (2e-5 v.s. 3e-5) and carefully designed learning rate decay strategies. This leads to a very interesting finding that a model may achieve a balance between learning new tasks and not forgetting old ones, only by controlling hyper-parameters like learning rate, although it is not clear whether the decoder or the encoder contribute more to the balance. However, in practice, we usually hope the model can learn as well as possible on new tasks while remembering old ones. Aligning with [d’Autumn19] and other previous efforts, if we set learning rate as 3e-5 to guarantee the model learning every new task at the best, it will inevitably make accuracy scores on previous tasks down to zero, a.k.a. catastrophic forgetting in the literature. In our work, we investigate a more practical but more tough scenario: whether a model still remember old tasks when it has been properly trained on new tasks. Our findings reveal that although a model seems to suffer from CF, its encoder component (BERT) can still generate distinguishable representing vectors for old tasks.

---

> > ### Author Response · Authors · 2022-11-10
> > **[Part 2] Response to Reviewer zUuP**
> >
> > 3. Probing study or direct evaluation on whole models
> >
> >     Intuitively, an encoder like BERT has billions of well-pre-trained parameters, whose gradients during finetuning on downstream tasks are tiny; but the decoder only contains tens of thousands of randomly-initialized parameters. As a result, when learning new tasks, its encoder and decoder may have different velocities of forgetting previous knowledge. In such cases, the poor scores directly evaluated on the whole model may NOT indicate its encoder completely loses the knowledge of previous tasks. It may be just because the decoder component cannot correctly decode the representations provided by encoder. To examine more exactly how much previous knowledge the encoder maintains, we can freeze the encoder and re-train a new decoder to guarantee the score merely relating to what knowledge the encoder memorizes, which falls into the convention of a probing study.
> >
> >     Here, we take an example to explain why direct evaluation on whole models may not be appropriate to measure an encoder’s ability to avoid forgetting. When comparing the bars of FT, EWC, ER in Figure 2(a) and Figure 3(a) in [Mehta21], we can see that three bars have the same length in Fig 2a but not in Fig 3a (separately grouped by ”-R” and ”-PT”). Among the overall accuracy scores of homogeneous tasks (Fig 2a), it shows FT=EWC=ER. But among the scores of diverse tasks (Fig 3a), the model faces a more serious shift of data distribution when switching to learn new tasks, which leads to a result of FT<EWC<ER. In other words, we argue that re-training a new decoder to examine old tasks can reduce the unexpected impact by the original decoder. So, in our probing study, an FT encoder will perform almost as well as a ER encoder, that is, if we only focus on the encoder part, bars in [Mehta21]’s Fig 3a should be equally high.
> >
> > 4. Different viewpoints to interpret the mechanism of CF.
> >
> >     We agree that a model without replay suffers from CF when evaluated on the whole model. But our probing study reveals that if we only consider the encoder part, when learning without replay, the BERT encoder can still maintain previous knowledge. To bridge the gap, we propose a new perspective to explain why a model can generate distinguishable representations for each task, but fails to perform well on old tasks. We argue that there are two levels in forgetting mechanism: intra-task forgetting and inter-task forgetting. In Section 4, we provide a set of toolkit to analyze the topological changes of BERT’s representing space. We confirm that BERT have an intrinsic ability to avoid intra-task forgetting. And we also analyze how Episodic Replay helps to alleviate inter-task forgetting (can be considered as the main reason of CF).
> >
> >     Our dichotomy between intra-task and inter-task forgetting, is inherited from the observation of our probing study, which separates the effects of the encoder and the decoder.
> >
> > 5. Values of our work.
> >
> >     Until reading this review, we realize that [Mehta21] has considered this intuitive but interesting topic earlier than us (unfortunately, we did not catch this work earlier.), and will revise our description on contributions accordingly. But, standing on those great efforts, we are still the first to reveal: **Even without any Episodic Replay or other LL algorithms, the pre-trained encoder of a model can maintain previous knowledge in a long term during task-incrementally learning, almost as effectively as the models enhanced by Episodic Replay**.
> >
> >     Additionally, [Mehta21] uses separate decoders for every task, and during training and evaluation, they purposely mask irrelevant decoders. In their work, an instance of Task 1 can never be predicted as Task 2. But we investigate a more tough scenario: The model can not access *task descriptor* and have to use a unified decoder. We confirm that even training under a unified decoder for all tasks, the encoder still keeps old knowledge without the help of Episodic Replay. Therefore, we can obtain a more precise conclusion that CF in task-incremental learning mainly comes from inter-task forgetting.
> >
> >     In this work, we do NOT aim to design a new lifelong learning algorithm or obtain SOTA scores on several benchmarks. What we want is to show our community these pre-trained models have the potentials that are not explored before. We hope this work can inspire people in future to consider the pre-trained encoder and the random-initialized decoder as two separate parts when designing new LL algorithms. And we expect there will be more works appearing to research how to take advantage of pre-trained encoders as much as possible.

---

> > > ### Author Response · Authors · 2022-11-10
> > > **[Part 3] Response to Reviewer zUuP**
> > >
> > > 6. Learning from [Mehta21].
> > >
> > >     We find the BERT encoder can refrain from forgetting by itself based on probing study, and we analyse it from the viewpoint of representing space. Due to the limitation of pages, we do not discuss where it comes from. [Mehta21] points out the pre-training process plays an important role. Their analysis inspires us to investigate whether there are other factors that can enhance a model to refrain from forgetting intrinsically. For example, the large-enough scale, the normalization between layers, or the self-attention mechanism.
> > >
> > >     [Mehta21] also provides experiments and analysis on CV benchmarks, although they only investigate text classification tasks. We examine the phenomenon to refrain from forgetting on more complex NLP tasks (Question Answering), which shows pre-trained language models can still keep old knowledge on tasks with more complex forms than classification. These results inspire us to study whether a pre-trained model can incrementally learn tasks with different forms (for example, text classification and semantic parsing, image classification and object detection).
> > >
> > >     [Mehta21] analyses the role of pre-training by comparing the loss contours. They find pre-training can make loss contours flatter, which leads to less forgetting. We think there can be certain connections between the loss contours and our representing cone toolkit. In our experiments, we also observe that the representing cones of a task in randomly-initialized BERT are less convergent than the cones in pre-trained BERT. It would be interesting to study whether we can build mathematical connections between loss analysis and the topological structure of representing space.
> > >
> > >     At the last, we sincerely thank reviewer zUuP to bring [Mehta21] to us. [Mehta21] and our work are motivated by the same intuition that well pre-trained models (like BERT) may have potentials to avoid forgetting. But we take different perspectives regarding the definition of forgetting and investigate under different scenarios, which makes us achieve different explanations w.r.t how pre-trained models have an intrinsic ability to reduce forgetting. And we further explain what is the actual reason causing CF (mainly due to inter-task forgetting but not intra-task) and how Episodic Replay can help alleviate inter-task forgetting. We think both the two works are valuable to our community, and can inspire future LL algorithms to concern about which part in a model is more possible to suffer from forgetting.
> > >
> > > Reference:
> > >
> > > **[d'Autumn19]** Cyprien de Masson d’Autume, Sebastian Ruder, Lingpeng Kong, and Dani Yogatama. **Episodic memory in lifelong language learning**. In *Proceedings of the 33rd International Conference on Neural Information Processing Systems*, pp. 13132–13141, 2019.
> > >
> > > **[Mehta21]** Sanket Vaibhav Mehta, Darshan Patil, Sarath Chandar, and Emma Strubell. **An empirical investigation of the role of pre-training in lifelong learning**. In *arXiv preprint*, arXiv:2112.09153, 2021.
> > >
> > > **[Tenny19]** Ian Tenney, Patrick Xia, Berlin Chen, Alex Wang, Adam Poliak, R. Thomas McCoy, Najoung Kim, Benjamin Van Durme, Sam Bowman, Dipanjan Das, and Ellie Pavlick. **What do you learn from context? Probing for sentence structure in contextualized word representations**. In *International Conference on Learning Representations*, 2019.
> > >
> > > **[Wang20]** Zirui Wang, Sanket Vaibhav Mehta, Barnabas Poczos, and Jaime G Carbonell. **Efficient meta lifelong-learning with limited memory**. In *Proceedings of the 2020 Conference on Empirical Methods in Natural Language Processing (EMNLP)*, pp. 535–548, 2020.

---

> > > > ### Comment · Reviewer_zUuP · 2022-12-04
> > > > **Appreciate the detailed response.**
> > > >
> > > > This is great, and I am glad you're working on incorporating this prior work. As I said, I like the paper other than the one issue that you're now working to resolve, so I will update my score.

---

> > > > > ### Author Response · Authors · 2022-12-05
> > > > > **Thanks to Reviewer zUuP**
> > > > >
> > > > > We again thank you for your comments and appreciation of our work. We will revise our paper based on these helpful suggestions, and add our new results in the final version.

---

### Decision · Program_Chairs · 2023-01-20

**Decision:**

Accept: poster

**Justification For Why Not Higher Score:**

See meta-review

**Justification For Why Not Lower Score:**

See meta-review

**Metareview: Summary, Strengths And Weaknesses:**

This paper explores whether BERT forgets representations of previous tasks over the course of being trained on new tasks. The method tracks the encoding ability of BERT for specific tasks before, during, and after learning new tasks. They find that BERT can actually refrain from forgetting when learning a sequence of tasks, contrary to existing studies about catastrophic forgetting. The authors believe this is due to the fact that BERT has a strong potential to produce high-quality representations for previous tasks even without memory replay

**Strengths:**

Overall, the reviewers seem to have positive opinions of the paper, with some praising the quality of the presentation and experiments and others noting the potential usefulness of the novel methods proposed.

**Weaknesses:**

There are some points of confusion or unclear presentation that the reviewers have raised, such as the lack of clarity around the definition of a "topological(ly) ordered" rotating process and the contradiction between some of the statements and results. It will be important for the authors to address these points in order to improve the clarity and coherence of the paper.


**Note From Pc:**

if the above contains the word "oral" or "spotlight" please see: "oral" presentation means -> notable-top-5% and "spotlight" means -> notable-top-25%. As stated in our emails, we are disassociating presentation type from AC recommendations

**Summary Of Ac-Reviewer Meeting:**

N/A